# Leveraging a Simulator for Learning Causal Representations from Post-Treatment Covariates for CATE

**Lokesh Nagalapatti**\*                                     *nlokeshiisc@gmail.com*
*Department of Computer Science and Engineering*
*IIT Bombay*

**Pranava Singhal**\*                                     *pranava.psinghal@gmail.com*
*Department of Computer Science*
*Stanford University*

**Avishek Ghosh**                                     *avishek_ghosh@iitb.ac.in*
*Department of Computer Science and Engineering*
*IIT Bombay*

**Sunita Sarawagi**                                     *sunita@iitb.ac.in*
*Department of Computer Science and Engineering*
*IIT Bombay*

**Reviewed on OpenReview:** *https://openreview.net/forum?id=vmmgFW3ztz*

## Abstract

Treatment effect estimation involves assessing the impact of different treatments on individual outcomes. Current methods estimate Conditional Average Treatment Effect (CATE) using observational datasets where covariates are collected before treatment assignment and outcomes are observed afterward, under assumptions like positivity and unconfoundedness. In this paper, we address a scenario where both covariates and outcomes are gathered after treatment. We show that post-treatment covariates render CATE unidentifiable, and recovering CATE requires learning treatment-independent causal representations. Prior work shows that such representations can be learned through contrastive learning if counterfactual supervision is available in observational data. However, since counterfactuals are rare, other works have explored using simulators that offer synthetic counterfactual supervision. Our goal in this paper is to systematically analyze the role of simulators in estimating CATE. We analyze the CATE error of several baselines and highlight their limitations. We then establish a generalization bound that characterizes the CATE error from jointly training on real and simulated distributions, as a function of the real-simulator mismatch. Finally, we introduce SimPONet, a novel method whose loss function is inspired from our generalization bound. We further show how SimPONet adjusts the simulator's influence on the learning objective based on the simulator's relevance to the CATE task. We experiment with various DGPs, by systematically varying the real-simulator distribution gap to evaluate *SimPONet*'s efficacy against state-of-the-art CATE baselines.

## 1 Introduction

In Conditional Average Treatment Effect (CATE) task, the goal is to estimate the difference in an outcome $Y$, as an individual $Z$ is subject to different treatments $T$. The gold standard to estimating such effects is Randomized Control Trials, which are often expensive, and with the easy availability of observational data, there is extensive interest in harnessing them for deriving these estimates. The first step in estimating treatment effects from observational datasets is to determine the set of covariates that, when conditioned upon, make the treatment effects

---

\* Lokesh and Pranava contributed equally. This work was conducted by Pranava during their tenure at IIT Bombay

identifiable. Prior works (Künzel et al., 2019; Nie & Wager, 2021; Curth & van der Schaar, 2023; Shalit et al., 2017; Nie et al., 2021; Shalit et al., 2016; Chauhan et al., 2023; Curth & van der Schaar, 2021; Zhang et al., 2020; Shi et al., 2019; Stuart, 2010), assume that such covariates are observed, and gathered prior to treatment, with outcomes $Y$ observed after the treatment is given. However, collecting such datasets is challenging as it requires tracking the same individuals over two distinct time points. As a result, readily available observational datasets can sometimes contain both covariates $X$ and the outcomes $Y$ recorded together post-treatment. $X, Y$ are observed for individuals characterized by a latent representation $Z$. For example, in economics, policymakers implement different taxation policies $T$ aimed at improving an outcomes $Y$ like Gross Domestic Expenditure of the individuals $Z$. To identify effective policies, they may rely on domain expertise or conduct small-scale RCTs where collecting pre-treatment covariates is feasible. However, the true effectiveness of a policy is only revealed through large-scale testing on the population after its implementation. Collecting such datasets typically involves obtaining post-treatment covariates $X$ and their associated outcomes $Y$ together (Ashenfelter, 1978; Angrist, 1995).

Even in other domains such as voluntary healthcare surveys, only post-treatment data about patients might be accessible. In medical imaging, an image taken under a specific instrument setting (treatment) may be evaluated to determine whether switching to a different setting would improve a subsequent diagnosis (outcome).

We present our setup in the top panel of Fig. 1 marked RealDGP (Data Generating Process for the real distribution), where the latent variables $Z$ causally produce the observed treatment $T$, outcome $Y$, and covariates $X$. We begin this paper by presenting an impossibility result in this context.

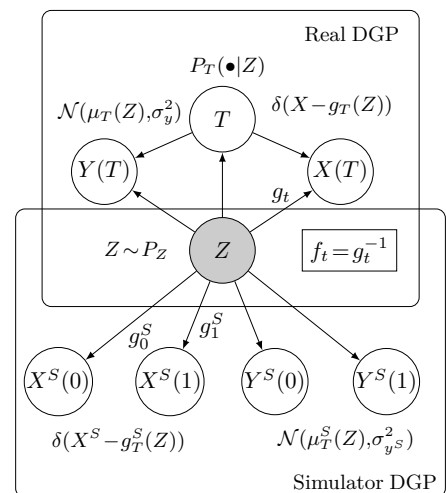

Figure 1: The Data Generating process for Real and Simulator.

**Lemma 1.** *The Conditional Average Treatment Effect (CATE) of $T$ on $Y$, given $X$, is not identifiable using i.i.d. samples of the observed variables from the true data-generating process depicted in the top panel of Fig. 1.*

*proof.* Since $X$ is a collider, conditioning on it opens the backdoor path $T \rightarrow X \leftarrow Z \rightarrow Y$. Furthermore, as $Z$ is latent, this backdoor path remains open, making CATE unidentifiable from $X, T$, and $Y$ alone (Pearl, 2015).

The main takeaway from the lemma is that certain additional assumptions are unavoidable for achieving identifiability. The lemma further emphasizes that the key to identifiability lies in extracting treatment-independent causal representations from the post-treatment $X$ that affect $Y$. One such assumption that allows for the recovery of causal representations is counterfactual supervision in real data. Prior work (Von Kügelgen et al., 2021) demonstrates that, under such an assumption, contrastive losses can be applied to pairs of covariates that differ by treatment to extract $Z$ from $X$ and $T$. While some works (Nagalapatti et al., 2022; Bachman et al., 2019) assume direct access to such counterfactual supervision in real data, others rely on simulators that generate high-quality synthetic counterfactuals (Von Kügelgen et al., 2021; Zimmermann et al., 2021). However, these are strong assumptions since counterfactuals are rarely available in real-world scenarios, and while simulators are more feasible, assessing their quality or relevance to the downstream task during training is challenging. Therefore, our goal in this paper is to leverage simulators only to the extent they remain relevant to the CATE task. We conduct a theoretical analysis to derive generalization bounds that show how CATE error worsens as the mismatch between real and simulated distributions increases. These insights motivate our proposed algorithm, *SimPONet*, which uses simulated data to apply regularizers inspired by our generalization bound. *SimPONet*'s aim is to enhance CATE estimates beyond what is achievable with observational data alone. Through experiments, we systematically vary the distributional gap between real and synthetic data across various DGPs, demonstrating that *SimPONet* consistently outperforms multiple baselines in estimating CATE.

**Contributions:** 1) We address Treatment Effect Estimation with post-treatment covariates —- a *non-identifiable* challenge – by leveraging a simulator that offers synthetic counterfactual supervision. 2) We assess the CATE errors for three baselines that can be trained on real/simulated data, and highlight their limitations. 3) Next, we consider a joint training framework, and derive a generalization bound that characterizes the

CATE error as a function of real-simulator distributional mismatch. 4) This analysis motivates our method, *SimPONet*, a novel algorithm that uses simulated samples to improve CATE estimates beyond what can be achieved from observational data alone. 5) To our knowledge, this is the *first* systematic study on the role of simulators in CATE estimation. 6) Experiments across various DGPs confirm *SimPONet*'s effectiveness.

## 2 Related Work

### 2.1 CATE with Pre-Treatment Covariates

The primary challenge addressed here is handling confounding that arises out of biased treatment assignment in observational datasets. The main ideas explored include: estimating pseudo-outcomes for missing treatments in the training dataset and then using these to train effect predictors (Gao & Han, 2020; Curth & van der Schaar, 2021; Nie & Wager, 2021; Kennedy, 2020; Yoon et al., 2018; Zhang et al., 2020; Nagalapatti et al., 2024a); adding targeted regularizers to ensure consistent ITE estimates (Shi et al., 2019; Nie et al., 2021; Zhang et al., 2022); learning balanced representation of covariates across treatment groups (Shalit et al., 2016; 2017; Yao et al., 2018; Chauhan et al., 2023; Wang et al., 2024; Wu et al., 2023); matching to near-by covariates (Stuart, 2010; Rosenbaum & Rubin, 1983; Iacus et al., 2012; Schwab et al., 2018; Kallus, 2020; Nagalapatti et al., 2024b); and weighing losses to mitigate confounding (Hassanpour & Greiner, 2019a;b; Jung et al., 2020; Ozery-Flato et al., 2018).

### 2.2 CATE with Post-Treatment Covariates

This is our setting, and is more challenging because it falls into the third rung (counterfactual) of Pearl's causal ladder (Pearl & Press, 2000). Please refer (Pearl, 2015) to for a formal proof. In economics, post-treatment variables in trials are known to exacerbate estimated causal effects (Coppock, 2019; HOMOLA et al., 2024; King, 2010). Post-treatment variables have been used to estimate selection bias $P(T|Z)$ in observational data (Bareinboim & Pearl, 2012; Bareinboim & Tian, 2015; Correa et al., 2018). A closely related work is (Huang et al., 2023) that leverages post-treatment variables for estimating treatment effects but differs from us since they assume: (1) covariates $X$ causally affect $Y$, and (2) an entangled version of $X,Z$ is observed; they simply focus on disentangling $Z$ through representation learning.

### 2.3 Real-World Applications of using Simulators for Estimating CATE

We provide two examples from medicine and electrochemistry to show how simulators aid CATE estimation in practice:

**Medicine.** Simulators play a crucial role in pharmacology, particularly for assessing drug efficacies. For instance, the SimBiology toolbox [1] in MATLAB is commonly used to predict the effects of `SGLT2` inhibitors ($T$) on type-2 diabetes ($Y$) while considering post-treatment covariates ($X$) such as plasma glucose levels, gut glucose levels, urinary glucose excretion, and liver insulin levels. SimBiology enables modeling these effects using differential and algebraic equations that are often calibrated on target populations to minimize the real-simulator mismatch. Despite not perfectly replicating reality, such simulators are invaluable for early-stage clinical trial decisions and have demonstrated utility in modeling short-term treatment effects (Dalla Man et al., 2007).

**Electrochemistry.** Another application involves recommending optimal electrode materials to maximize battery capacity ($Y$). By observing $Y$ under various electrode materials ($T$) and post-treatment variables like charge/discharge rate, internal resistance, and temperature distribution ($X$), the Ansys Battery Cell and Electrode Simulator [2] provides electrochemical simulations. This tool has been used by Volkswagen Motorsport for comprehensive multiphysics simulations to design and validate battery models. Such simulators are highly relevant for practical decision-making in industries.

These examples illustrate the practical relevance of simulators across different fields. While simulators cannot fully replace real data or randomized controlled trials (RCTs), they offer valuable insights that can reduce the number of RCTs needed for optimal treatment identification. Our paper aims to characterize the CATE

---

[1] https://in.mathworks.com/videos/series/simbiology-tutorials-for-qsp-pbpk-and-pk-pd-modeling-and-analysis.html

[2] https://www.ansys.com/applications/battery/battery-cell-and-electrode

error when using imperfect simulators in conjunction with real observational data. Additionally, *SimPONet* maximizes the utility of simulators by leveraging the highly correlated simulator's treatment effects with real-world effects, without relying on the exact correlation of individual potential outcomes.

## 3 Problem Formulation

We use random variables $X, T, Y$ to denote post-treatment covariates, binary treatments, and outcomes respectively. The observational dataset has $n$ samples: $D_{\text{trn}} = \{(\boldsymbol{x}_i, t_i, y_i)\}_{i=1}^n$ where $t_i \in \mathcal{T} = \{0, 1\}$ denotes treatment, $\boldsymbol{x}_i \in \mathcal{X} \subset \mathbb{R}^{n_x}$ denotes covariates observed after $t_i$ is applied, and $y_i$ the resulting outcome. We use the Neyman-Rubin potential outcomes framework to denote $Y_i(t), X_i(t)$ as the potential outcome and covariate for unit $i$ under a treatment $t$. The main challenge is the absence of counterfactuals in $D_{\text{trn}}$, i.e., for each unit $i$, we observe covariates and outcomes under only one treatment $t_i$.

We use the random variable $Z \in \mathcal{Z} \subset \mathbb{R}^{n_z}$ to denote the causal representations of covariates $X$. $Z$ generates $X$ via treatment-specific covariate generating functions $g_t : \mathcal{Z} \mapsto \mathcal{X}$ for $t \in \{0, 1\}$. We assume that $g_t$ is diffeomorphic (Locatello et al., 2019a;b; Von Kügelgen et al., 2021); i.e., it is smooth, invertible, and has a smooth inverse. The assumption that $g_t$ is diffeomorphic is required to establish the theoretical identifiability of $\tau$. A non-invertible transformation risks losing information when mapping $Z, T$ to $X$. If this lost information includes features necessary for identifiability, then $\tau$ becomes unidentifiable from the observed $X, T, Y$. Diffeomorphism ensures that all factors involved in generating $X$ are preserved within $X$ so that there exists inverse functions $f_t : \mathcal{X} \mapsto \mathcal{Z}, \forall t \in \{0, 1\}$ that could recover the causal representations $Z$ back. In Sec. 5.6, we experiment with non-invertible transformations and find that both the baselines and *SimPONet* maintain robust performance, even when these assumptions are violated in practice.

A sample is obtained from the real DGP as follows: (1) $z_i \sim P_Z$, (2) $t_i \sim P(T|z_i)$, (3) $\boldsymbol{x}_i \sim P(X|z_i, t_i) = \delta(X - g_{t_i}(z_i))$, where $\delta$ denotes the dirac-delta distribution, (4) $y_i \sim P(Y|z_i, t_i) = \mathcal{N}(\mu_{t_i}(z_i), \sigma_y^2)$ is sampled from a Gaussian with mean $\mu_{t_i}(z_i)$ and constant variance $\sigma_y^2$. Here, $\mu_t : \mathcal{Z} \mapsto \mathcal{Y}$ $\forall t$ generates responses for individuals with latent representations $z$ under treatment $t$. We express the *factual* observed outcome for $i$ as $Y_i(t_i) = \mu_{t_i}(f_{t_i}(\boldsymbol{x}_i))$, and the missing *counterfactual* (CF) outcome under $1 - t_i$ as $Y_i(1 - t_i) = \mu_{1-t_i}(f_{t_i}(\boldsymbol{x}_i))$.

**Our Goal** is Conditional Average Treatment Effect (CATE) estimation which quantifies the difference in outcomes due to a change in treatment. Given a test unit $(\boldsymbol{x}_j, t_j)$, its CATE is given by $\tau_j = \mathbb{E}[Y_j(T = 1) - Y_j(T = 0)|\boldsymbol{x}_j, t_j]$. As argued earlier, estimating $\tau$ from post-treatment data involves the sub goal of learning causal representations of observed covariates $X$ using a function $f_t : X \mapsto Z$. We use $\tau : \mathcal{Z} \mapsto \mathcal{Y}$ to express the treatment effect using the latent $z_j$ as $\tau(z_j) = \mu_1(z_j) - \mu_0(z_j)$. Since $f_t$ inverts $X$ to give $Z$, the same effect can also be expressed for $(\boldsymbol{x}_j, t_j)$ using $\tau_X$ as $\tau_X(\boldsymbol{x}_j, t_j) = \mu_1(f_{t_j}(\boldsymbol{x}_j)) - \mu_0(f_{t_j}(\boldsymbol{x}_j))$ where $\tau_X : \mathcal{X} \times \mathcal{T} \mapsto \mathcal{Y}$. Notice that $\tau_X(\bullet, t) = \tau \circ f_t(\bullet)$. When estimating $\tau_X$, the factual outcome is easy, all we need to do is fit a regression model on the observation data. The main challenge lies in estimating the counterfactual outcome under treatment $1 - t_j$.

Theorem 1 in (Locatello et al., 2019a) presents an impossibility result stating that $f_t$ which maps covariates $X$ to their treatment-independent causal representations $Z$ is *not* identifiable solely using $D_{\text{trn}}$. The main hurdle is that multiple DGPs can yield the same marginal distribution $P(X, T)$, making it impossible to isolate the true DGP. However, prior work has shown how to learn $f_t$ with *counterfactuals*, requiring that $D_{\text{trn}}$ includes both covariates $X_i(t_i)$ and $X_i(1 - t_i)$. Theorem 4.4 of (Von Kügelgen et al., 2021) shows that such counterfactual supervision allows for recovery of $Z$ up to a diffeomorphic transformation $h$ using contrastive learning. Proposition 2 in (Zimmermann et al., 2021) further shows that $h$ is, in particular, a rotation in an $n_z$ dimensional unit-normalized hypersphere. While some prior works assume direct access to counterfactual supervision in real data (Nagalapatti et al., 2022; Bachman et al., 2019), others rely on high-quality synthetic counterfactuals from simulators (Xie, 2018; Kaur et al., 2021). In contrast, our approach seeks to leverage simulators only to the extent that they improve the downstream CATE task. We next formally define the simulator's data generating process.

**Simulator DGP** The simulator generates paired instances giving rise to a counterfactual dataset $D_{\text{syn}} = \{\boldsymbol{x}_i^S(0), \boldsymbol{x}_i^S(1), y_i^S(0), y_i^S(1)\}$ generated using the DGP as shown in the lower panel in Figure 1. The simulated instances are obtained as follows: (1) $z_i \sim P_Z$; i.e., $Z$ is sampled from the same distribution as real, (2) post-treatment covariates $\boldsymbol{x}_i^S(t) \sim P(X^S|Z = z_i, T = t) = \delta(X^S - g_t^S(z_i))$ under *both* treatments $t = \{0, 1\}$.

$g_t^S : \mathcal{Z} \mapsto \mathcal{X}$ $\forall t$ are diffeomorphic functions, and (3) corresponding outcomes $y_i^S(0)$, $y_i^S(1)$ are sampled from $P(Y^S | Z = z_i, T = t) = \mathcal{N}(\mu_t^S(z_i), \sigma_{y^S}^2)$, where $\mu_t^S : \mathcal{Z} \mapsto \mathcal{Y}, \forall t$. Note that $z_i$ remains hidden even in $D_{\text{syn}}$. We use "$S$" in the superscript to indicate a simulator component. Now we describe some metrics that assess the distance between real and simulator DGP.

**Definition 1** $[d_{\boldsymbol{x}|t}(f_t, f_t^S)]$ We assess the distance between the real and synthetic causal representation extractors $f_t$ and $f_t^S$ using the following expected distance: $d_{\boldsymbol{x}|t}(f_t, f_t^S) = \mathbb{E}_{\boldsymbol{x} \sim P(X|t)}\left[||f_t(\boldsymbol{x}) - f_t^S(\boldsymbol{x})||_2^2\right]$.

**Definition 2** $[d_z(\tau, \tau^S)]$ We assess the distance between the real and simulator CATE functions on the $P_Z$ distribution as: $d_z(\tau, \tau^S) = \mathbb{E}_{z \sim P_Z}\left[(\tau(z) - \tau^S(z))^2\right]$. Under composition with a diffeomorphic function $h$, we write $d_h(\tau, \tau^S) = \mathbb{E}_{z \sim P_Z}\left[(\tau(h(z)) - \tau^S(h(z)))^2\right]$.

**Assumptions for Identifying CATE $\tau_X$.** We summarise the assumptions that are needed on the real dataset $D_{\text{trn}}$ and simulated counterfactual dataset $D_{\text{syn}}$ to identify the CATE function $\tau_X$: (A1) *Positivity:* $P(T = t | Z = z) > 0, \ \forall t \in \mathcal{T}, z \in \mathcal{Z}$. (A2) *Diffeomorphic Covariate Generation:* Covariates in both real and synthetic distributions are obtained through diffeomorphic transformations of $Z$ under any treatment $T$. (A3) *Identifiability of $\tau$ given $Z$:* The causal factors $Z$ that generate $X$ form a sufficient adjustment set, blocking backdoor paths between $T$ and $Y$, thus making $\tau$ identifiable from $Z$. Note that A2 and A3 together ensure that $X$ contains information about all the relevant latent factors that affect the outcome $Y$ and is a weaker notion of the commonly used *unconfoundedness* assumption.

**CATE Error ($\mathcal{E}_{\textbf{CATE}}$).** Given a test dataset $D_{\text{tst}} = \{(\boldsymbol{x}_j, t_j, y_j(0), y_j(1))\}_{j=1}^m$, with each $\boldsymbol{x}_j$ rendered under $t_j$, we compute the empirical error incurred in estimating CATE using mean squared error as $\mathcal{E}_{\text{CATE}} = \frac{1}{m} \sum_{j \in D_{\text{tst}}} [\tau_j - \widehat{\tau}_j]^2$ where $\tau_j = y_j(1) - y_j(0)$ is the true effect and $\widehat{\tau}_j$ is the predicted effect for the instance $(\boldsymbol{x}_j, t_j)$.

The CATE error in general can be decomposed across treatment $T$ as

$$\mathcal{E}_{\text{CATE}} = \sum_{t \in \mathcal{T}} P(T = t) \, \mathcal{E}_{\text{CATE}}^t \ \text{ where } \mathcal{E}_{\text{CATE}}^t = \int_{\boldsymbol{x} \in \mathcal{X}} [\tau_X(\boldsymbol{x}, t) - \widehat{\tau_X}(\boldsymbol{x}, t)]^2 P(\boldsymbol{x}|t) d\boldsymbol{x}$$

**Definition 3.** Let us define *factual* error $\mathcal{E}_F^t$ and *counterfactual* error $\mathcal{E}_{CF}^t$ on samples with observed treatment $t$ and missing treatment $1 - t$ as follows:

$$\mathcal{E}_F^t = \int_{\boldsymbol{x} \in \mathcal{X}} [\mu_t(f_t(\boldsymbol{x})) - \widehat{\mu}_t(\widehat{f}_t(\boldsymbol{x}))]^2 P(\boldsymbol{x}|t) d\boldsymbol{x} \text{ and } \mathcal{E}_{CF}^t = \int_{\boldsymbol{x} \in \mathcal{X}} [\mu_{1-t}(f_t(\boldsymbol{x})) - \widehat{\mu}_{1-t}(\widehat{f}_t(\boldsymbol{x}))]^2 P(\boldsymbol{x}|t) d\boldsymbol{x}$$

**Lemma 2.** *The CATE error is related to the factual and counterfactual error as:* $\mathcal{E}_{CATE}^t \leq 2\mathcal{E}_F^t + 2\mathcal{E}_{CF}^t$ *[Proof in Appendix A.5.1]*

# 4 Learning Causal Representations for CATE

Our task involves learning four functions: $\widehat{f}_t$ that extracts the causal representations from $X(t)$ and $\widehat{\mu}_t$ that estimates the outcomes $Y(t)$ for $t \in \{0, 1\}$. With access to *counterfactual simulated* data $D_{\text{syn}}$ and *observational real* data $D_{\text{trn}}$, one can come up with the following approaches for estimating CATE: 1) SimOnly, which only uses $D_{\text{syn}}$, and 2) RealOnly which only uses $D_{\text{trn}}$ to estimate $\mu_t$, (3) Real$_\mu$Sim$_f$, which uses $D_{\text{syn}}$ to estimate $f_t$ and subsequently, $D_{\text{trn}}$ to estimate $\mu_t$. We now discuss the training approach for each of these methods, delve into their shortcomings, and then present our proposed method *SimPONet*.

To illustrate the shortcomings, we consider a test instance $\boldsymbol{x}^\star$ generated under treatment $T = 1$ (without loss of generality) and derive the CATE error expression for it in the population setting, as $|D_{\text{trn}}| \to \infty$ and $|D_{\text{syn}}| \to \infty$.

## 4.1 The SimOnly Estimator

SimOnly solely uses $D_{\text{syn}}$. It leverages the counterfactual supervision provided by the simulator and identifies the simulator's DGP as follows:

(Step 1) Estimate the synthetic causal representation extractor $f^S$ from covariate pairs $\{\boldsymbol{x}_i^S(0), \boldsymbol{x}_i^S(1)\}$ using contrastive learning Von Kügelgen et al. (2021):

$$\{\widetilde{f}_0^S, \widetilde{f}_1^S\} = \underset{\{\widehat{f}_0^S, \widehat{f}_1^S\}}{\operatorname{argmin}} \sum_{i=1}^{|D_{\mathrm{syn}}|} \left[ -\log \frac{\exp(\mathrm{sim}(\hat{z}_i(1), \hat{z}_i(0)))}{\sum_{j \neq i} \sum_{t,t'} \exp(\mathrm{sim}(\hat{z}_i(t), \hat{z}_j(t')))} \right] \text{ where } \hat{z}_i(t) = \widehat{f}_t^S(\boldsymbol{x}_i^S(t)) \tag{1}$$

where $\mathrm{sim}(\bullet, \bullet)$ is cosine similarity, $(\boldsymbol{x}_i^S(t), \boldsymbol{x}_i^S(1-t))$ denotes a positive pair with the same underlying latent $z_i$. A negative pair $(\boldsymbol{x}_i^S(t), \boldsymbol{x}_j^S(t'))$ has different $(z_i, z_j)$. Contrastive learning increases similarity of representations of positive pairs $(\hat{z}_i(0), \hat{z}_i(1))$ while pushing apart the negative pairs $(\hat{z}_i(t), \hat{z}_{j \neq i}(t'))$.

**Lemma 3.** *As $|D_{syn}| \to \infty$, contrastive training with paired counterfactual covariates as shown in Eq. 1 recovers $\widetilde{f}_t^S = h \circ f_t^S$ where $h$ is a diffeomorphic transformation. Moreover, when the latent space $\mathcal{Z} \subset \mathbb{S}^{(n_z-1)}$ (unit-norm hypersphere in $\mathbb{R}^{n_z}$), $h$ is a rotation transform by Extended Mazur-Ulam Theorem as shown in (Zimmermann et al., 2021) (Proposition 2).*

*[Refer Appendix A.5.2 for more details.]*

The main insight from the above lemma is that, given counterfactual supervision, it is possible to recover causal representations $Z$ from post-treatment covariates $X$ up to a rotation $h$, making CATE identifiable in the simulated distribution, as we demonstrate below.

(Step 2) Estimate $\widetilde{\tau}^S(z) = \widetilde{\mu}_1^S(z) - \widetilde{\mu}_0^S(z)$ with supervision on difference of outcomes $\tau^S(f_t^S(\boldsymbol{x}_i^S(t))) = y_i^S(1) - y_i^S(0)$ as

$$\widetilde{\tau}^S = \underset{\widehat{\tau^S}}{\operatorname{argmin}} \sum_{\boldsymbol{x}^S \in D_{\mathrm{syn}}} \left[ \tau^S(f_t^S(\boldsymbol{x}^S(t))) - \widehat{\tau^S}(\widetilde{f}_t^S(\boldsymbol{x}^S(t))) \right]^2 \tag{2}$$

The SimOnly method uses these estimates as-is on real data, i.e. $\widehat{\tau} = \widetilde{\tau}^S$ and $\widehat{f}_t = \widetilde{f}_t^S, \ \forall t \in \mathcal{T}$. We analyze below the error incurred with such estimates on real data.

**CATE error:** In the population setting, since $\widetilde{f}_t^S = h \circ f_t^S$, we see that the optimization problem in Eq. 2 yields $\widetilde{\tau}^S = \tau^S \circ h^{-1}$ as its solution. Thus, for an instance $\boldsymbol{x}^\star$ from the real distribution under treatment 1, the true CATE is $\tau(f_1(\boldsymbol{x}^\star))$. The CATE error using SimOnly becomes:

$$\left[ \tau(f_1(\boldsymbol{x}^\star)) - \tau^S \circ h^{-1}(h \circ f_1^S(\boldsymbol{x}^\star)) \right]^2 = \left[ \tau(f_1(\boldsymbol{x}^\star)) - \tau^S(f_1^S(\boldsymbol{x}^\star)) \right]^2.$$

This shows that for SimOnly to provide accurate CATE estimates, the simulator must perfectly align with the real world; i.e., $\tau^S = \tau$ and $f_t = f_t^S$ for all $t$. However, designing such simulators is highly challenging in practice, making this method unsuitable for CATE.

## 4.2 The RealOnly Estimator

RealOnly solely uses real observational data $D_{\mathrm{trn}}$. Since $D_{\mathrm{trn}}$ lacks counterfactual covariates, this model cannot apply contrastive training and therefore cannot explicitly supervise the recovery of causal representations. Instead, it focuses on regressing the factual outcomes $y_i(t_i)$ from post-treatment covariates $\boldsymbol{x}_i(t_i)$. In terms of the four learning parameters, its learning objective is:

$$\underset{\{\widehat{\mu}_0, \widehat{\mu}_1, \widehat{f}_0, \widehat{f}_1\}}{\operatorname{argmin}} \sum_{i=1}^{|D_{\mathrm{trn}}|} (y_i - \widehat{\mu}_{t_i}(\widehat{f}_{t_i}(\boldsymbol{x}_i)))^2$$

However, since $\widehat{\mu}_t, \widehat{f}_t$ are not individually supervised, we might as well collapse them into a composition $\mu_t^F = \mu_t \circ f_t$; yielding $\mu_{t_i}^F(\boldsymbol{x}_i) = y_i$, and thereby CATE as $\widehat{\tau_X}(\boldsymbol{x}, t) = \widehat{\mu}_1^F(\boldsymbol{x}) - \widehat{\mu}_0^F(\boldsymbol{x})$.

RealOnly is consistent in estimating the factual outcomes, because as $|D_{\mathrm{trn}}| \to \infty$, we have $\widehat{\mu}_t^F = \operatorname{argmin}_{\widehat{\mu}_t^F} \mathbb{E}_{\boldsymbol{x} \sim P(\boldsymbol{x}|t)} \left[ \left( \widehat{\mu}_t^F(\boldsymbol{x}) - \mu_t^F(\boldsymbol{x}) \right)^2 \right] = \mu_t^F$ and therefore, the factual error $\mathcal{E}_F^t = 0$. However, RealOnly

incurs a significant error when estimating the counterfactual outcome, which in turn contributes to the CATE error, as shown below.

**CATE error:** The true CATE for $\boldsymbol{x}^\star$ obtained using treatment 1 can be written as $\tau(f_1(\boldsymbol{x}^\star)) = \mu_1(f_1(\boldsymbol{x}^\star)) - \mu_0(f_1(\boldsymbol{x}^\star))$. Then, the CATE error for RealOnly is computed as:

$$\left[\left(\mu_1(f_1(\boldsymbol{x}^\star)) - \mu_0(f_1(\boldsymbol{x}^\star))\right) - \left(\widehat{\mu}_1^F(\boldsymbol{x}^\star) - \widehat{\mu}_0^F(\boldsymbol{x}^\star)\right)\right]^2 = \left[\left(\mu_1(f_1(\boldsymbol{x}^\star)) - \widehat{\mu}_1^F(\boldsymbol{x}^\star)\right) - \left(\mu_0(f_1(\boldsymbol{x}^\star)) - \widehat{\mu}_0^F(\boldsymbol{x}^\star)\right)\right]^2$$

In the population setting, using the consistency of factual estimates, the CATE error reduces to $\left[\mu_0(f_1(\boldsymbol{x}^\star)) - \mu_0^F(\boldsymbol{x}^\star)\right]^2$. This error is zero when $f_1(\boldsymbol{x}^\star) = f_0(\boldsymbol{x}^\star)$. Thus, for RealOnly to provide accurate CATE estimates, the treatment must not affect the post-treatment covariates, i.e., $g_0(z) = g_1(z)\ \forall z$ in which case their inverse are equal $f_0 = f_1$. However, this assumption is often unrealistic. For instance, in pharmacology, different drugs typically induce distinct effects on patient covariates, limiting the applicability of this model in such settings.

**Remark:** The post-treatment covariates $X$ can be viewed as a special case of the pre-treatment covariates $Z$ when the covariate-generating functions $g_0 = g_1$. In such cases, our proposed *SimPONet* algorithm offers no distinct advantage, and existing CATE methods designed for pre-treatment covariates suffice and should be used instead.

### 4.3 The Real$_\mu$Sim$_f$ Estimator

Unlike SimOnly, which uses $D_{\text{syn}}$ to learn both $\widehat{f}_t$ and $\widehat{\mu}_t$, this approach leverages $D_{\text{syn}}$ solely to learn the representation extractor $\widehat{f}_t$. Specifically, it assumes that $\widehat{f}_t = \widetilde{f}_t^S$, as obtained from Eq. 1. Thereafter, it learns the $\widehat{\mu}_t$ parameters by applying a factual loss on $D_{\text{trn}}$ to estimate

$$\widehat{\mu}_0, \widehat{\mu}_1 = \underset{\{\widehat{\mu}_0, \widehat{\mu}_1\}}{\operatorname{argmin}} \sum_{D_{\text{trn}}} (y_i - \widehat{\mu}_{t_i}(\widetilde{f}_{t_i}^S(\boldsymbol{x}_i)))^2$$

We call this method Real$_\mu$Sim$_f$ since it learns the outcome parameters $\mu$ from real samples while learning representation extractor $f_t$ from the simulator.

**CATE error:** One condition under which the Real$_\mu$Sim$_f$ model achieves zero CATE error is when $\widetilde{f}_t^S = f_t$ for each treatment $t$. This requires that the simulator aligns with real-world covariates, specifically $\boldsymbol{x}_t = g_t(z) = g_t^S(z) = \boldsymbol{x}_t^S$. This limitation arises because the model learns the representation extractor solely from $D_{\text{syn}}$, without making adjustments for real covariates.

In summary, we described three possible CATE estimators and showed that each method would provide accurate CATE estimates under certain strong assumptions about the real and simulator DGPs. Given that none of these assumptions would hold in practice, we now turn to exploring a joint training framework that learns simultaneously from both real and simulated samples.

### 4.4 The *SimPONet* Estimator

We first conduct a theoretical analysis to derive a generalization bound that characterizes the CATE error as a function of the mismatch between the real and simulator distributions. This analysis forms the basis for our proposed method, *SimPONet*, whose loss function is inspired by the bound.

**Lemma 4.** *Assume $\tau$ is $K_\tau$-Lipschitz, and $\widetilde{f}^S$ and $\widetilde{\tau}^S$ are estimates from the simulator DGP obtained from the optimization in Eq. 1, 2. Then, the CATE error on the estimates $\widehat{f}_t$ and $\widehat{\tau}$ admits the following bound:*

$$\mathcal{E}_{CATE}^t(\widehat{f}_t, \widehat{\tau}) \leq [8\mathcal{E}_F^t + 12d_h(\widehat{\tau}, \widetilde{\tau}^S) + 12K_\tau^2 d_{\boldsymbol{x}|t}(\widehat{f}_t, \widetilde{f}_t^S)] + [12d_z(\tau, \tau^S) + 12K_\tau^2 d_{\boldsymbol{x}|t}(f_t, f_t^S)]$$

*where $d_{\boldsymbol{x}|t}, d_z, d_{h(z)}$ are distance functions in Sec. 3 and $\mathcal{E}_F^t$ is the factual loss.   [Proof in Appendix A.5.2.]*

The expressions in blue are constants that capture the discrepancy between real and simulated distributions and cannot be minimized. In contrast, the remaining terms can be minimized by training on $D_{\text{trn}}$ and $D_{\text{syn}}$. As $|D_{\text{trn}}|$ approaches infinity, the factual error $\mathcal{E}_F^t$ can be made to approach zero, while the other minimizable distance terms act as regularizers. The term $d_h(\widehat{\tau}, \widetilde{\tau}^S)$ can assist in regularizing the outcome parameters $\widehat{\mu}_t$,

whereas $d_{\boldsymbol{x}|t}(\widehat{f}_t, \widetilde{f}_t^S)$ can aid in regularizing the parameters of the causal representation extractor functions $\widehat{f}_t$. This analysis leads to our proposed approach *SimPONet* whose overall loss is as follows:

$$
\min_{\{\widehat{\mu}_t, \widehat{f}_t\}} \underbrace{\sum_{D_{\mathrm{trn}}} \left(y_i - \widehat{\mu}_{t_i}(\widehat{f}_{t_i}(\boldsymbol{x}_i))\right)^2}_{\text{Factual Loss on } D_{\mathrm{trn}}} + \lambda_f \underbrace{\sum_{D_{\mathrm{trn}}} \|\widetilde{f}_{t_i}^S(\boldsymbol{x}_i) - \widehat{f}_{t_i}(\boldsymbol{x}_i)\|_2^2}_{d(\widetilde{f}_t^S, \widehat{f}_t) \text{ regularizer}} + \lambda_\tau \underbrace{\sum_{D_{\mathrm{syn}}} \sum_{t \in \{0,1\}} \left(\tau_i^S - \widehat{\tau}(\widetilde{f}_t^S(\boldsymbol{x}_i^S(t)))\right)^2}_{\tau^S \text{ regularizer on } D_{\mathrm{syn}}}
$$

(3)

where $\tau_i^S = y_i^S(1) - y_i^S(0)$ and $\lambda_\tau, \lambda_f > 0$ are loss weights. $\widehat{\tau}(\bullet) = \widehat{\mu}_1(\bullet) - \widehat{\mu}_0(\bullet)$ denotes the estimated CATE.

*SimPONet* relaxes the strict equality $\widehat{f}_t = \widetilde{f}_t^S$ used by $\text{Real}_\mu \text{Sim}_f$, and instead uses $\widetilde{f}_t^S$ as a regularizer, while ensuring that $\widehat{\mu}_t$ accurately predicts the factual outcomes for instances in $D_{\mathrm{trn}}$. It also imposes the $\tau^S$ loss on simulated instances to leverage any potential similarity between the true treatment effect, $\tau$, and the simulated treatment effect, $\tau^S$. Furthermore, the $\tau^S$ loss is essential to prevent degenerate solutions that would cause *SimPONet* to collapse to the $\text{Real}_\mu \text{Sim}_f$ estimator. This is because applying regularization solely on $\widehat{f}_t$ can drive the regularizer $\|\widehat{f}_t(\boldsymbol{x}) - \widetilde{f}_t^S(\boldsymbol{x})\|_2^2$ to zero, leading to $\widehat{f}_t = \widetilde{f}_t^S$, while still minimizing the factual error $\mathcal{E}_F^t$ by updating $\widehat{\mu}_t$ accordingly. Consequently, *SimPONet* would collapse into the $\text{Real}_\mu \text{Sim}_f$ estimator, making the $\tau^S$ loss critical in avoiding such degeneracies.

**Adjusting Loss Weights.** *SimPONet* adjusts the loss weight $\lambda_f$ for learning $\widehat{f}_t$ by comparing the *factual errors* of the RealOnly model, which trains on $X$, with those of the $\text{Real}_\mu \text{Sim}_f$ model, trained using simulated causal representations $\widetilde{f}_t^S(\boldsymbol{x})$. If RealOnly consistently outperforms $\text{Real}_\mu \text{Sim}_f$ in factual error, we infer that the simulated representations may not generalize well to the real distribution, prompting *SimPONet* to reduce $\lambda_f$. By default, $\lambda_f$ is set to 1; however, if $\text{Real}_\mu \text{Sim}_f$ exhibits a notably higher factual error, *SimPONet* lowers $\lambda_f$ to $10^{-4}$.

In contrast, tuning $\lambda_\tau$ requires $\tau$ supervision on real data, which is unavailable. Prior work (Curth & van der Schaar, 2021; Nagalapatti et al., 2024b; Künzel et al., 2019) argue that while outcome functions $\mu_t$ can be complex, the difference function $\tau = \mu_1 - \mu_0$ is often simpler. For instance, if we consider $\mu_t^S = \mu_t + c$ (with $c > 0$), we can make the factual outcomes to diverge arbitrarily while their corresponding $\tau$ and $\tau^S$ remain equal. Therefore, while comparing the factual errors between SimOnly and RealOnly models to set $\lambda_\tau$ is appealing, it maybe a poor choice in practice. So, *SimPONet* always sets $\lambda_\tau$ to its default 1.

We present the *SimPONet*'s pseudocode in Appendix A.4.

## 5 Experiments

We conduct experiments that are designed to address the following research questions:

RQ1 How do different methods compare with varying discrepancies between real and simulator in settings with closed form estimates; i.e., without errors due to finite-sample training?

RQ2 How does *SimPONet* compare to other SOTA baselines that assume pre-treatment covariates?

RQ3 What are the contributions of individual loss terms in *SimPONet*?

RQ4 How does *SimPONet* fare against the baselines when $\mu_t$ exhibits complex non-linear behavior?

RQ5 How do CATE methods perform when trained directly on (a) post-treatment covariates $X$, or (b) pre-treatment $Z$, or (c) simulated causal representations $\widetilde{f}^S$ from Eq. 1?

### 5.1 Neural Architecture and Hyperparameters

For the Linear experiments in RQ1, we omit shared layers in Fig. 5, and set $\widehat{f}_0$ and $\widehat{f}_1$ as $n_x \times n_x$ matrices, and $\widehat{\mu}_0$ and $\widehat{\mu}_1$ as $n_x \times 1$ vectors. For the real-world experiments in RQ2, with both $f_t, \mu_t$ as non-linear functions, we set $\widehat{\mu}_0$ and $\widehat{\mu}_1$ as 2-layer MLPs with hidden layers of 100 and 50 neurons. The shared layers have 2 hidden

layers with 50 and $n_x$ neurons, respectively. We impose the $d(\widehat{f}_t, \widetilde{f}_t^S)$ loss for *SimPONet* on outputs of the shared layers. For experiments in RQ4, we omit the shared layers while setting $\widehat{f}_t$ as linear layer. But since $\mu$ is non-linear, we use an MLP with one hidden layer of 50 neurons and ReLU activations for each $\widehat{\mu}_t$.

**Hyperparameters:** We implemented all baselines and *SimPONet* using `jax` within CATENets (Curth et al., 2021), a standard library for benchmarking state-of-the-art CATE estimation methods. To ensure consistency, we used the same MLP architecture, learning rates, optimizers, and other hyperparameters as the default settings in CATENets for baseline approaches. The unique hyperparameters for *SimPONet* are the loss weights $\lambda_f$ and $\lambda_\tau$. As described in Sec. 4.4, *SimPONet* tunes $\lambda_f$ by comparing the factual errors of the RealOnly and $\text{Real}_\mu\text{Sim}_f$ models, while $\lambda_\tau$ is always set to 1. CATENets applies early stopping based on factual error in the validation dataset, a common practice in CATE training. To ensure a fair comparison, we maintained consistent training and validation splits across all methods.

**Assessing Statistical Significance:** For CATE experiments, standard deviation is sometimes misleading to comment on the statistical significance of empirical results as noted in (Curth et al., 2021). So, for all experiments, we conduct a one-sided paired *t*-test with *SimPONet* as the baseline and enclose *p-values in brackets* to indicate statistical significance. Lower *p*-values favor *SimPONet*.

### 5.2   RQ1: Assessing Baselines Under Settings Without Finite Training Sample Errors

To address RQ1, we consider a setting where both the real and simulator DGPs as shown in Fig. 1 are linear. In particular, we generate the training datasets $D_{\text{trn}}$ and $D_{\text{syn}}$ as follows: (1) Latent variables $z \in \mathbb{R}^{n_z}$ are sampled from distribution $P_Z$. (2) Real and simulated covariates for a treatment $t$ are computed as $g_t(z) = z\boldsymbol{R}_t$ and $g_t^S(z) = z\boldsymbol{S}_t$, where $\boldsymbol{R}_t$ and $\boldsymbol{S}_t$ are invertible matrices. (3) Outcomes are generated as $\mu_t(z) = z^\top w_t$ and $\mu_t^S(z) = z^\top w_t^S$, where $w_t$ and $w_t^S$ are vectors in $\mathbb{R}^{n_z}$. We consider two datasets for RQ1: (1) Synthetic-Gaussian and (2) Real World-IHDP, differing in how $Z$ is obtained. In setting (1), $z \in \mathbb{R}^{10}$ is sampled from a standard Gaussian $\mathcal{N}(0,1)$, while for (2), $Z$ is taken from the real-world IHDP dataset (Appendix A.7) as-is.

Now, to systematically control the real-simulator mismatch, we need means to vary the following distances: $d(\boldsymbol{R}_0^{-1}, \boldsymbol{R}_1^{-1})$, $d(\boldsymbol{R}_t^{-1}, \boldsymbol{S}_t^{-1})$, and $d(\tau, \tau^S)$. We achieve this as follows: (1) Initialize $\boldsymbol{R}_0^{-1}, w_0 \sim \mathcal{N}(0,1)$. (2) To inject a distance $\gamma_R \in (0, 0.5)$ between $\boldsymbol{R}_0^{-1}$ and $\boldsymbol{R}_1^{-1}$, set $\boldsymbol{R}_1^{-1} = (1 - \gamma_R)\boldsymbol{R}_0^{-1} + \gamma_R\mathcal{N}(0,1)$. (3) Set $w_1 \sim \gamma w_0 + (1-\gamma)\mathcal{N}(0,1)$. We use $\gamma = 0.4$ in all experiments. (4) Similarly, inject a $\gamma_{RS}$ gap between $\boldsymbol{R}_t^{-1}$ and $\boldsymbol{S}_t^{-1}$. (5) For treatment effect parameters $w_\tau = w_1 - w_0$ in the real DGP, we sample its simulator counterpart with a gap $\gamma_\tau$ as $w_\tau^S = (1 - \gamma_\tau)w_\tau + \gamma_\tau\mathcal{N}(0,1)$ and set $w_t^S$ accordingly.

In linear settings, the optimization problems for the CATE estimators SimOnly, RealOnly, and $\text{Real}_\mu\text{Sim}_f$ admit closed-form solutions. We show the closed-form solutions in Table 7 in the Appendix, and a detailed derivation in Appendix A.6.

For *SimPONet*, a closed-form solution is not possible, so we solve it to a local optimum using alternating minimization over $\widehat{\mu}_t$ and $\widehat{f}_t$, with each alternating update in closed-form. We show the *SimPONet*'s update equations in Appendix A.6.4. In summary, the setting of RQ1 allows study of the impact of varying discrepancies between the real and simulator distributions without approximation errors due to finite training samples.

We show the results comparing *SimPONet* with the three baselines in Table 1[3] where we observe: (a) Across both synthetic and real-world settings, *SimPONet* achieves either the best or second-best performance. The CATE error for *SimPONet* remains controlled primarily due to its ability to bound errors in the counterfactual distribution. (b) In contrast, the RealOnly and $\text{Real}_\mu\text{Sim}_f$ models perform well only under certain restrictive conditions favorable to them, providing zero error on the factual distribution but very high counterfactual error, leading to poor CATE estimates.

### 5.3   RQ2: Comparing *SimPONet* with State-of-the-art CATE Baselines

We conduct experiments using semi-synthetic observational datasets commonly used to assess efficacy of treatment effect estimation methods in the literature: the Infant Health Development Program (IHDP) and

---

[3]Each entry in the table reports the CATE Error alongside its corresponding *p*-value as CATE Error (*p*-value). The *p*-value denotes the statistical significance of the hypothesis that *SimPONet* outperforms the baseline. We use a one-sided paired *t*-test for this assessment. Smaller *p*-values indicate stronger evidence in favor of *SimPONet*. We show the **best performing method** in green, and the second best method in yellow across all our tables.

Table 1: RQ1: In a linear DGP setting, we vary the gaps using $\gamma_R$ for $d(f_0, f_1)$ in the first column, $\gamma_{RS}$ for $d(f_t, f_t^S)$, and $\gamma_\tau$ for $d(\tau, \tau^S)$. "low" refers to 0.1 that simulates small distance, while "high" refers to 0.4. We run all experiments with five different seeds and report $p$-values of comparing the mean performance in bracket.

| $d(f_0, f_1)$ | $d(f_t, f_t^S)$ | $d(\tau, \tau^S)$ | Synthetic-Gaussian | | | | Real World-IHDP | | | |
|---|---|---|---|---|---|---|---|---|---|---|
| | | | SimOnly | RealOnly | $\text{Real}_\mu\text{Sim}_f$ | *SimPONet* | SimOnly | RealOnly | $\text{Real}_\mu\text{Sim}_f$ | *SimPONet* |
| 0.00 | high | high | 2.82 (0.27) | **0.00 (1.00)** | 15.75 (0.01) | 2.58 (0.00) | 3.57 (0.11) | **0.00 (1.00)** | 48.76 (0.05) | 3.20 (0.00) |
| low | low | low | 0.63 (0.00) | 2.47 (0.02) | 1.19 (0.01) | **0.54 (0.00)** | 1.00 (0.44) | 3.43 (0.02) | 2.73 (0.00) | **0.97 (0.00)** |
| low | low | high | 1.57 (0.16) | 2.47 (0.08) | **1.19 (0.83)** | 1.39 (0.00) | 1.62 (0.26) | 3.43 (0.04) | 2.73 (0.02) | **1.49 (0.00)** |
| low | high | low | 2.14 (0.22) | 2.47 (0.00) | 15.75 (0.01) | **1.85 (0.00)** | 3.67 (0.31) | 3.43 (0.48) | 48.76 (0.05) | **3.37 (0.00)** |
| low | high | high | 2.82 (0.26) | **2.47 (0.56)** | 15.75 (0.01) | 2.57 (0.00) | 3.57 (0.11) | 3.43 (0.39) | 48.76 (0.05) | **3.19 (0.00)** |
| high | low | low | 0.63 (0.00) | 13.86 (0.02) | 1.19 (0.01) | **0.54 (0.00)** | 1.00 (0.47) | 47.78 (0.06) | 2.73 (0.00) | **0.98 (0.00)** |
| high | low | high | 1.57 (0.16) | 13.86 (0.03) | **1.19 (0.83)** | 1.39 (0.00) | 1.62 (0.27) | 47.78 (0.06) | 2.73 (0.02) | **1.50 (0.00)** |
| high | high | low | 2.14 (0.21) | 13.86 (0.03) | 15.75 (0.01) | **1.85 (0.00)** | 3.67 (0.31) | 47.78 (0.06) | 48.76 (0.05) | **3.38 (0.00)** |
| high | high | high | 2.82 (0.26) | 13.86 (0.04) | 15.75 (0.01) | **2.57 (0.00)** | 3.57 (0.11) | 47.78 (0.06) | 48.76 (0.05) | **3.19 (0.00)** |

the Atlantic Causal Inference Conference (ACIC) datasets. These datasets contain real-world *pre-treatment* covariates ($Z$). Please refer Appendix A.7 for more details on these datasets.

To align these datasets with our study, we apply RealNVP Normalizing Flows (Dinh et al., 2016) to transform pre-treatment covariates $Z$ into post-treatment $X$. Flows are non-linear deep neural networks that ensure invertibility of the covariate generating functions $g_t, g_t^S$. We consider randomly initialized flows with two coupling layers. We used the flows $g_0, g_1$ on real data, and two other distinct flows $g_0^S, g_1^S$ to obtain covariates in synthetic data from $Z$. We borrow the real outcomes as-is from the ground truth dataset. However, we synthesize simulator outcomes with a gap of $\gamma_\tau$ as follows: (1) sample $w_\tau^S \in \mathbb{R}^{n_z} \sim \mathcal{N}(0,1)$ (2) set $\tau^S(z) = \tau(z) + (\sigma(\tau) \cdot \gamma_\tau \cdot z^\top w_\tau^S)$, where $\sigma(\tau)$ is the standard deviation of CATE labels in the real dataset. Scaling by $\sigma(\tau)$ ensured comparability between $\tau$ and $\tau^S$. Thus, when $\gamma_\tau = 0$, $\tau = \tau^S$; when $\gamma_\tau = 1$, $\tau$ is disparate from $\tau^S$.

We evaluated *SimPONet* against various baselines from the well-known CATENets (Curth et al., 2021), a benchmarking library for CATE estimation. Since the baseline methods are not designed to extract the causal representations, we provided them with representations extracted by simulated causal representation extractor $\widetilde{f}_t^S(\boldsymbol{x})$ as input for a fair comparison. Running these baselines with post-treatment covariates $X$ directly as input yielded much poorer results as shown in Fig. 3. We also compared *SimPONet* with SimOnly, RealOnly, and $\text{Real}_\mu\text{Sim}_f$ baselines that we developed in our theoretical analysis. We present the results in Table 2 for $\gamma_\tau = 0.1$, and defer the results for larger $\gamma_\tau$ to Appendix 5.7. We make the following key observations:

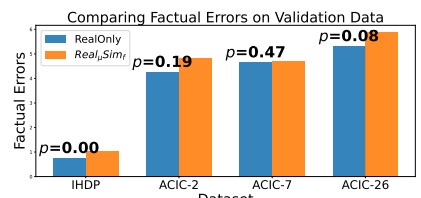

Figure 2: Factual errors with $p$-values shown above bars. For IHDP, RealOnly consistently outperforms $\text{Real}_\mu\text{Sim}_f$.

(a) **IHDP**: This dataset contains 25 pre-treatment covariates, 19 of which are binary. Contrastive training struggled to capture these binary features, causing $\text{Real}_\mu\text{Sim}_f$, which uses $\widetilde{f}_t^S(\boldsymbol{x})$ as input, to consistently underperform RealOnly, which directly uses $\boldsymbol{x}$. As shown in Fig 2, the $p$-value is zero for the IHDP dataset, but significantly larger for the others. As a result, we set the weight of the regularizer $d(\widehat{f}_t, \widetilde{f}_t^S)$, controlled by $\lambda_f$, to 1e-4 for IHDP, while keeping $\lambda_f$ at its default value of 1 for ACIC. Overall, *SimPONet* achieved the best performance.

(b) **ACIC-2:** This dataset is unique in that the true CATE, $\tau$, is constant across all individuals in the observational data, implying that its standard deviation $\sigma(\tau) = 0$ for real samples. As a result, our approach to synthesizing the simulated CATE, $\tau^S$, given by $\tau^S(z) = \tau(z) + (\sigma(\tau) \cdot \gamma_\tau \cdot z^\top w_\tau^S)$, yields $\tau^S(z) = \tau(z)$ for all $z$. This leads to perfect alignment between the synthetic and true CATE, causing SimOnly to outperform all other methods on this dataset. Although *SimPONet* could have improved by assigning a higher weight to the $\tau^S$ regularizer, tuning this weight would typically require supervision on $\tau$, which we avoid.

Table 2: RQ2: Comparison of *SimPONet* with several pre-treatment baselines and post-treatment proposals. *p*-values for paired t-tests against *SimPONet* are in brackets. Lower *p*-values indicate statistical significance. *SimPONet* outperforms others overall, while SimOnly performs best on ACIC-2 since $\tau = \tau^S$.

| Method | IHDP | ACIC-2 | ACIC-7 | ACIC-26 | ACIC-All |
|---|---|---|---|---|---|
| RNet (Nie & Wager, 2021) | 1.54 (0.00) | 3.30 (0.00) | 5.91 (0.04) | 6.06 (0.18) | 5.78 (0.03) |
| XNet (Künzel et al., 2019) | 1.0 (0.00) | 0.43 (0.15) | 5.49 (0.17) | 5.1 (0.38) | 4.45 (0.41) |
| DRNet (Schwab et al., 2020) | 0.96 (0.00) | 0.24 (0.59) | 5.53 (0.15) | 5.08 (0.39) | 4.45 (0.40) |
| CFRNet (Shalit et al., 2017) | 0.96 (0.00) | 0.36 (0.26) | 5.55 (0.15) | 5.09 (0.38) | 4.67 (0.32) |
| FlexTENet (Curth & van der Schaar, 2021) | 0.96 (0.00) | 0.32 (0.32) | 5.46 (0.19) | 5.04 (0.40) | 4.85 (0.37) |
| DragonNet (Shi et al., 2019) | 0.96 (0.00) | 0.29 (0.41) | 5.57 (0.14) | 5.09 (0.38) | 4.60 (0.40) |
| IPW (Robins et al., 1994) | 0.96 (0.00) | 0.36 (0.24) | 5.56 (0.15) | 5.09 (0.38) | 4.45 (0.40) |
| $k$-NN (Stuart, 2010) | 0.96 (0.00) | 0.33 (0.33) | 5.48 (0.18) | 5.13 (0.37) | 4.44 (0.40) |
| PerfectMatch (Schwab et al., 2018) | 0.98 (0.00) | 0.56 (0.11) | 5.75 (0.08) | 5.13 (0.37) | 4.56 (0.28) |
| StableCFR (Wu et al., 2023) | 1.01 (0.00) | 1.09 (0.03) | 5.56 (0.15) | 5.08 (0.43) | 4.82 (0.14) |
| ESCFR (Wang et al., 2024) | 0.96 (0.00) | 0.27 (0.47) | 5.55 (0.15) | 5.79 (0.21) | 4.73 (0.18) |
| PairNet (Nagalapatti et al., 2024b) | 0.97 (0.00) | 0.12 (0.85) | 5.46 (0.23) | 5.05 (0.37) | 4.44 (0.41) |
| SimOnly | 0.94 (0.00) | **0.00 (0.98)** | 6.65 (0.00) | 6.60 (0.12) | 6.45 (0.02) |
| RealOnly | 0.83 (0.13) | 11.23 (0.01) | 14.81 (0.05) | 8.18 (0.01) | 9.82 (0.00) |
| $\mathrm{Real}_\mu\mathrm{Sim}_f$ | 0.96 (0.00) | 0.17 (0.76) | 5.57 (0.14) | 5.09 (0.38) | 4.52 (0.38) |
| *SimPONet* | **0.79 (0.00)** | 0.26 (0.00) | **5.04 (0.00)** | **4.67 (0.00)** | **4.36 (0.00)** |

Table 3: RQ3: Impact of regularizers. Here, $-d(\widehat{f}_t, \widetilde{f}_t^S)$ represents our loss 3 with $\lambda_f = 0$, and $-\tau^S$ means $\lambda_\tau = 0$. A negative value implies *SimPONet* with all regularizers outperforms the ablation where one regularizer is disabled.

| | | | IHDP - Linear $f_t$, Linear $\mu_t$ | | GP - Linear $f_t$, Non-Linear $\mu_t$ | |
|---|---|---|---|---|---|---|
| $d(f_0, f_1)$ | $d(f_t, f_t^S)$ | $d(\tau, \tau^S)$ | $-d(\widehat{f}_t, \widetilde{f}^S)$ | $-\tau^S$ | $-d(\widehat{f}_t, \widetilde{f}^S)$ | $-\tau^S$ |
| 0.00 | high | high | +1.29 (1.00) | -1.07 (0.18) | -0.55 (0.31) | -0.62 (0.26) |
| high | low | low | -0.64 (0.04) | +0.01 (0.51) | -0.29 (0.24) | -0.04 (0.47) |
| high | low | high | -0.40 (0.11) | +0.00 (0.50) | -0.42 (0.10) | -0.19 (0.34) |
| high | high | low | +1.74 (0.99) | -0.03 (0.48) | -0.66 (0.27) | -0.71 (0.25) |
| high | high | high | +1.30 (1.00) | -0.03 (0.45) | -0.72 (0.21) | -0.97 (0.18) |

(c) **ACIC-7 and ACIC-26:** The CATENets baselines significantly outperform the RealOnly model, because of high-quality causal representations extracted by $\widetilde{f}_t^S$. This is demonstrated by $\mathrm{Real}_\mu\mathrm{Sim}_f$ outperforming RealOnly on factual error (see Fig. 2). In ACIC-7 and ACIC-26, *SimPONet* achieves the best results by leveraging the closeness between $\tau$ and $\tau^S$. FlexTENet (Curth & van der Schaar, 2021), which shares parameters between $\widehat{\mu}_0$ and $\widehat{\mu}_1$, and PairNet (Nagalapatti et al., 2024b), which applies losses on pairs of close-by samples are strong contenders to *SimPONet*.

**ACIC-All:** The last column in the table presents results across all 77 seeds of the ACIC dataset. *SimPONet* achieves the lowest mean CATE error overall. However, for certain seeds, *SimPONet*'s CATE error remains comparable to baselines such as PairNet, $k$-NN, as indicated by them having *p*-values close to 0.4. The RealOnly model consistently performs worse across all seeds due to the covariate functions $g_0$ and $g_1$ producing disparate post-treatment covariates $X_0$ and $X_1$ when applied to $Z$. Similarly, the SimOnly model suffers due to the distributional mismatch between the real and simulator data.

### 5.4 RQ3: Ablation of *SimPONet* Losses

We evaluate the impact of the $d(\widehat{f}_t, \widetilde{f}_t^S)$ and $\tau^S$ regularizers in *SimPONet*'s objective 3. We experiment with the Linear IHDP dataset (Sec. 5.2) and the Non-Linear Gaussian process dataset (Sec. 5.5). For the Linear IHDP dataset, in the $-\tau^S$ case, we added an $L_2$ penalty on $w_t$ for the alternating minimization to work. Table 3 presents the *difference* in CATE errors of *SimPONet* and the ablation, averaged over five seeds with *p*-values. A negative entry means *SimPONet* does better than the ablation. We observe that $\tau^S$ loss is very effective since *SimPONet* outperforms the $-\tau^S$ in both datasets. For IHDP, removing $d(\widehat{f}_t, \widetilde{f}_t^S)$ loss helps. We could not set $\lambda_f$ weight to be small because both RealOnly and $\mathrm{Real}_\mu\mathrm{Sim}_f$ achieved zero factual error. Despite this,

Table 4: RQ4: Results for linear covariate and GP-based nonlinear outcome functions. We run each experiment 5 times and show the $p$-values. *SimPONet* outperforms others in many settings. RealOnly is a strong contender.

| $d(f_0,f_1)$ | $d(f_t,f_t^S)$ | $d(\tau,\tau^S)$ | SimOnly | RealOnly | $\text{Real}_\mu\text{Sim}_f$ | *SimPONet* |
|---|---|---|---|---|---|---|
| 0.00 | high | high | 5.30 (0.09) | 2.99 (0.11) | 3.17 (0.09) | **1.94 (0.00)** |
| low | low | low | 1.77 (0.15) | 1.06 (0.73) | **1.05 (0.77)** | 1.26 (0.00) |
| low | low | high | 1.56 (0.08) | **0.98 (0.65)** | 1.07 (0.44) | 1.05 (0.00) |
| low | high | low | 4.60 (0.04) | 2.87 (0.07) | 3.12 (0.04) | **1.88 (0.00)** |
| low | high | high | 4.15 (0.04) | 2.96 (0.02) | 3.11 (0.01) | **1.60 (0.00)** |
| high | low | low | 2.39 (0.14) | 1.19 (0.68) | **1.12 (0.77)** | 1.37 (0.00) |
| high | low | high | 1.94 (0.11) | **0.98 (0.83)** | 1.16 (0.58) | 1.21 (0.00) |
| high | high | low | 7.42 (0.09) | 2.65 (0.38) | 2.95 (0.25) | **2.37 (0.00)** |
| high | high | high | 5.70 (0.07) | 2.80 (0.17) | 3.16 (0.09) | **2.10 (0.00)** |

*SimPONet* with both regularizers comfortably outperformed the other proposals demonstrating it as a better candidate for our task.

### 5.5 RQ4: Linear covariate function $g$ and non-linear outcome function $\mu$

Now, we consider a more complex setup where the covariate functions $g_t$ and $g_t^S$ remain linear, but the outcome functions $\mu_t$ and $\mu_t^S$ are nonlinear in $Z$. In particular, we sample the outcomes $y$ and $y^S$ using Gaussian Processes (GPs) (Rasmussen & Williams, 2005). Let $GP(0,\mathcal{K}_\gamma)$ denote a GP with an RBF kernel of width $\gamma$, so a higher $\gamma$ results in a more complex function. To sample the $\mu_0,\mu_1$ such that their difference $\tau$ has a gap $\gamma$, we follow: (1) Sample $\tau$ using a GP: $\tau \sim GP(0,\mathcal{K}_\gamma)$. (2) Sample $\mu_0 \sim GP(0,\mathcal{K}_1)$. (3) Set $\mu_1 \sim \mu_0 + \tau$. As before, we set $\gamma = 0.4$. Now, to sample $\mu_0^S,\mu_1^S$ such that $d(\tau,\tau^S) = \gamma_\tau$: (1) Set $\tau^S \sim \tau + GP(0,\mathcal{K}_{\gamma_\tau})$. (2) Sample $y_0^S \sim GP(0,\mathcal{K}_1)$. (3) Set $y_1^S = y_0^S + \tau^S$.

We estimate $\widetilde{f}^S$ in Eq. 1 in closed-form, whereas we learn other parameters using gradient descent. We show the results in Table 4 where we observe: (a) *SimPONet* outperforms the other baselines in five out of nine settings (b) In alignment with our theory, $\text{Real}_\mu\text{Sim}_f$ performs better than others only when $d(f_t,f_t^S)$ is small. In summary, when the properties of the underlying DGP are unclear, *SimPONet* proves to be an effective approach for learning $\tau$.

### 5.6 RQ5: Comparing CATE methods when trained on $Z$ vs $X$ vs $\widetilde{f}^S(X)$

To address RQ5, we use the IHDP dataset to evaluate the baseline models when trained on post-treatment covariates $X$ directly, and we compare these results with those from Table 2, where the baselines were trained on simulated causal representations, $\widetilde{f}_t^S(\boldsymbol{x})$. We also evaluate the baselines trained on pre-treatment $Z$ to explicitly show the detrimental impact of post-treatment covariates on the CATE error.

In addition, we extend Table 2 by considering non-invertible, non-linear transformations with Multi-Layer Perceptrons (MLPs) on the IHDP dataset to assess the robustness of *SimPONet* and baselines when the diffeomorphism assumption is violated. To this end, we used two-layer MLPs for $g_t$ and $g_t^S$ to generate covariates from $Z$. We present the results in Fig. 3, and make the following observations:

In the pre-treatment setting, causal representation extraction is unnecessary, making simulator supervision inconsequential; thus, we omit the four post-treatment CATE methods introduced in this paper. For other baselines, Fig. 3 shows that they perform significantly worse with post-treatment $X$ than with pre-treatment $Z$, underscoring the importance of causal representation recovery. DragonNet achieved the best performance with pre-treatment covariates.

For Normalizing Flow-generated covariates, CATE error consistently decreased when baselines used simulator-based causal representations, $\tilde{S}_t(X)$, rather than $X$, validating the utility of simulators in extracting causal representations. However, the CATE error remains significantly higher compared to using pre-treatment covariates $Z$, highlighting that simulators, while helpful, are not ideal and exhibit a distributional gap between real and simulator distributions. This gap impacts all CATE methods. Notably, *SimPONet*, specifically

designed to leverage simulator supervision and incorporate regularizers using simulator-generated data, achieves the best results.

In the MLP-based covariate experiments with non-invertible covariate functions $g_0$, $g_1$, *SimPONet* consistently outperformed all baselines, exhibiting trends similar to those observed with Normalizing Flow-generated covariates. This suggests that the diffeomorphism assumption, while necessary for our theoretical results, may be inconsequential in practice.

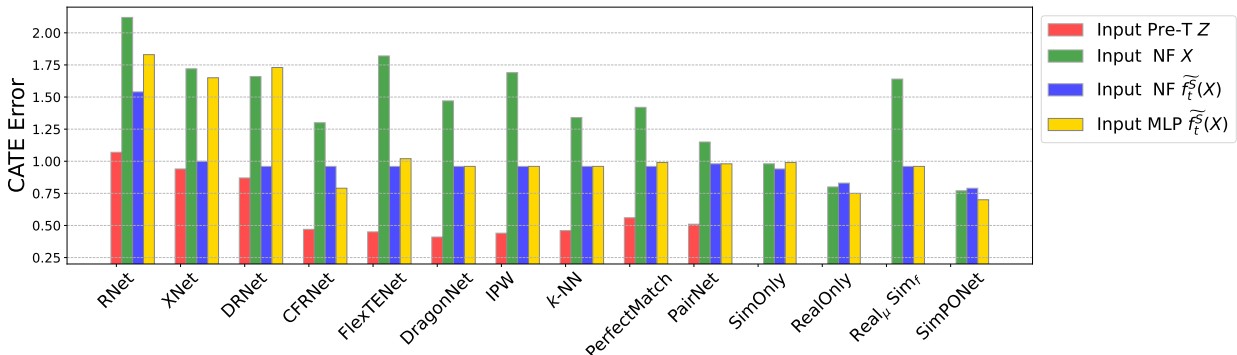

Figure 3: Comparing CATE errors under pre-treatment $Z$, and MLP, Normalizing flow generated post-treatment covariates $X$.

## 5.7   Varying $\gamma_\tau$ in Arbitrary DGP Experiment

The results of the varying $\gamma_\tau$ experiment are presented in Fig. 4, where we compare two approaches that leverage simulator data during training: SimOnly and *SimPONet*. Across all $\gamma_\tau$ gaps, we observe that *SimPONet* consistently outperforms SimOnly in three out of four datasets, with ACIC-2 being the exception. This is expected, as ACIC-2 satisfies the condition $\tau_s = \tau$. The performance gains of *SimPONet* are particularly notable in the ACIC-7 and ACIC-26 datasets, where the CATE error for SimOnly escalates significantly at larger $\gamma_\tau$ values. The exact error values for these cases are shown in the inset subplot at the top-right. These findings underscore our argument: while SimOnly can perform well on simulators closely aligned with the real world, it struggles with real-world simulators that diverge from reality. In contrast, *SimPONet*'s adjustment strategies—enabled by theoretically grounded regularizers derived from the CATE error analysis—yield much more reliable CATE estimates.

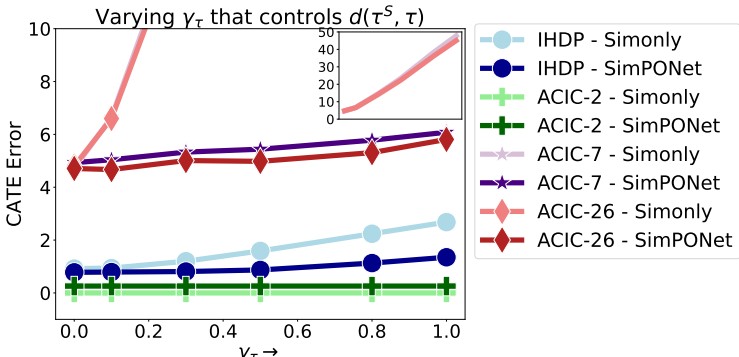

Figure 4: We vary $\gamma_\tau$, which controls the gap between the synthetic CATE, $\tau^S$, and the real CATE, $\tau$. Each dataset is represented by a distinct color, where the pale version of the color indicates SimOnly and the darker version denotes *SimPONet*. For ACIC-7 and ACIC-26, as $\gamma_\tau$ increases, the CATE error grows significantly. Therefore, we present these results as an inset figure in the top-right corner.

Table 5: RQ5: Evaluating $\mathrm{Real}_\mu \mathrm{Sim}_f$ performance with $Z$ from $SimPONet$'s $\widehat{f}_t$ vs. $\widetilde{f}_t^S$ obtained from $D_{\mathrm{syn}}$. $\mathrm{Real}_\mu \mathrm{Sim}_f$ with $SimPONet$'s $\widehat{f}_t$ significantly outperforms in many settings indicating $SimPONet$ extracts better $Z$.

| | | | IHDP - Linear $f_t$, Linear $\mu_t$ | | GP - Linear $f_t$, Non-Linear $\mu_t$ | |
|---|---|---|---|---|---|---|
| $d(f_0,f_1)$ | $d(f_t,f_t^S)$ | $d(\tau,\tau^S)$ | $\mathrm{Real}_\mu\mathrm{Sim}_f$ | $\mathrm{Real}_\mu SimPONet_f$ | $\mathrm{Real}_\mu\mathrm{Sim}_f$ | $\mathrm{Real}_\mu SimPONet_f$ |
| 0.00 | high | high | 48.76 (0.05) | **3.28 (0.00)** | 3.17 (0.32) | **2.67 (0.00)** |
| low | low | low | 2.73 (0.00) | **0.97 (0.00)** | **1.05 (0.81)** | 1.55 (0.00) |
| low | low | high | 2.73 (0.02) | **1.49 (0.00)** | **1.07 (0.78)** | 1.55 (0.00) |
| low | high | low | 48.76 (0.05) | **3.48 (0.00)** | 3.12 (0.23) | **2.53 (0.00)** |
| low | high | high | 48.76 (0.05) | **3.21 (0.00)** | 3.11 (0.34) | **2.73 (0.00)** |
| high | low | low | 2.73 (0.00) | **0.98 (0.00)** | **1.12 (0.72)** | 1.29 (0.00) |
| high | low | high | 2.73 (0.02) | **1.49 (0.00)** | **1.16 (0.76)** | 1.42 (0.00) |
| high | high | low | 48.76 (0.05) | **3.37 (0.00)** | **2.95 (0.57)** | 3.11 (0.00) |
| high | high | high | 48.76 (0.05) | **3.23 (0.00)** | 3.16 (0.42) | **2.97 (0.00)** |

## 5.8 On the Quality of $Z$ Extracted by *SimPONet*

In this experiment, we evaluate the quality of representations learned by $SimPONet$. If $SimPONet$'s regularizers in the losses enable the recovery of representations identifiable for CATE, then a RealOnly model trained on these representations should outperform the $\mathrm{Real}_\mu\mathrm{Sim}_f$ model trained on representations extracted using contrastive training on the simulator dataset. We focus on the Linear IHDP and non-linear GP datasets, as they provide closed-form solutions for Objective 1, and and gives us a fine grained control to set the gaps between real and simulated datasets and assess the performance across many settings of these gaps. Specifically, we train the RealOnly model using $Z$ extracted from $\widehat{f}_t$ via $SimPONet$, and call it as $\mathrm{Real}_\mu SimPONet_f$, and compare it to $\mathrm{Real}_\mu\mathrm{Sim}_f$ trained from $\widetilde{f}^S$. Results in Table 5 show that $\mathrm{Real}_\mu SimPONet_f$ significantly outperforms the baseline $\mathrm{Real}_\mu\mathrm{Sim}_f$ across many DGP settings, with particularly strong results in the Linear IHDP case where $p$-values are consistently small. This demonstrates that $SimPONet$ effectively learns high-quality representations $Z$.

## 5.9 Sensitivity to Loss Weights

Table 6: This table assesses the sensitivity of the loss weights on all three versions of the IHDP dataset considered in our work – Synthetic Linear Sec. 5.2, Synthetic Non-Linear Sec. 5.5, and Semi-synthetic Sec. 5.3. We consider nine different settings of the loss weights in $SimPONet$'s objective 3. The table shows mean $\pm$ standard deviation of the CATE Error.

| $\lambda_\mu$ | $\lambda_\phi$ | Synthetic Linear | Synthetic Non-Linear | Semi-Synthetic |
|---|---|---|---|---|
| 0.1 | 0.1 | 3.46±0.91 | 4.23±2.21 | 0.72±0.19 |
| 0.1 | 0.5 | 3.42±0.84 | 3.55±1.21 | 0.70±0.19 |
| 0.1 | 1.0 | 3.43±0.78 | 3.40±1.15 | 0.71±0.20 |
| 0.5 | 0.1 | 3.41±0.90 | 4.06±2.21 | 0.74±0.19 |
| 0.5 | 0.5 | 3.38±0.88 | 3.32±1.06 | 0.73±0.19 |
| 0.5 | 1.0 | 3.37±0.82 | 3.23±1.00 | 0.71±0.20 |
| 1.0 | 0.1 | 3.38±0.86 | 3.99±2.17 | 0.71±0.20 |
| 1.0 | 0.5 | 3.39±0.87 | 3.24±1.02 | 0.72±0.19 |
| 1.0 | 1.0 | 3.38±0.85 | 3.14±0.93 | 0.71±0.21 |

In this experiment, we analyze the sensitivity of $SimPONet$'s performance to the loss weights $\lambda_f$ and $\lambda_\tau$ in its objective function. Using the IHDP covariates, we performed this analysis in the three variants used in our study. For the synthetic datasets, we set the real simulator gaps, as specified in Tables 1 and 4, to `high`. In the Semi-Synthetic IHDP setting, we evaluate performance on five randomly selected dataset versions. The results, summarized in Table 6, report the mean CATE error along with its standard deviation across dataset seeds. All methods consistently performed poorly when the loss weights are small. This indicates that regularizers are important to impose on post-treatment covariates. Across other settings, overall, we observe that $SimPONet$ is sensitive to the choice of hyperparameter settings in non-linear and semi-synthetic versions, and remains

fairly robust in the linear setting. However, tuning these hyperparameters is challenging in practice due to the absence of counterfactual data in real observational datasets. Despite this sensitivity, *SimPONet* outperforms achieves either comparable performance or manages to surpass the baseline methods across different loss weight configurations.

## 6 Conclusion

This paper addressed the challenge of estimating treatment effects from post-treatment covariates a setting not identifiable from observational data alone. We proposed to tackle this task using off-the-shelf simulators that synthesize counterfactuals, in contrast to prior work that relied on real-world counterfactuals, which limited their practical applicability. Our theoretical analysis established a bound on the CATE error based on the distributional mismatch between real and simulated data. Notable, ours is the first work to systematically analyze the role of simulators in CATE estimation. We introduced *SimPONet*, a framework that jointly learns from real and simulated samples to enhance CATE estimates beyond what could be achieved from observational data alone. Extensive experiments across various DGPs demonstrated that *SimPONet* is a robust and effective method for estimating CATE from post-treatment data.

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

# A Appendix / supplemental material

## A.1 Code

We have released the code in the URL `https://github.com/nlokeshiisc/simponet_release`.

## A.2 Learning Counterfactual Simulators

Here we discuss prior works that train generative models for synthesizing counterfactuals. In general, to obtain counterfactuals in the real distribution, we need to follow three steps (Pawlowski et al., 2020): (a) *abduction*, inverting $X$ to obtain $Z$, (b) *action*, applying a new treatment, and (c) *prediction* generating a new $X$ under the new treatment. These steps require prior knowledge of the DGP specifications, which are often difficult to define and cannot be learned from observational data alone (Pawlowski et al., 2020). Consequently, many methods bypass the principled approach and use pre-trained models like Diffusion models and Large Language models to generate pseudo counterfactuals from a related synthetic domain. Such simulators are proposed across various modalities, including images (Pawlowski et al., 2020; Gu et al., 2023; Thiagarajan et al., 2021; Pan & Bareinboim, 2024; Sauer & Geiger, 2021; Jeanneret et al., 2022), text (Madaan et al., 2020; Calderon et al., 2022; Chen et al., 2023; Dixit et al., 2022; Robeer et al., 2021), and healthcare (Lang et al., 2023; Wang et al., 2020; Dhinagar et al., 2024). Prior research (Gondal et al., 2019) shows that while such simulated data is not directly usable for downstream tasks, they provide strong inductive biases that transfer well to the real distribution. Our method can incorporate any such counterfactual generators as simulators, provided they contribute to learning causal representations that are predictive of CATE.

## A.3 *SimPONet* Architecture

We present an overview of the *SimPONet* model architecture in Figure 5. Our model has four primary parameters: $\widehat{f}_0$ and $\widehat{f}_1$ for extracting causal representations, and $\widehat{\mu}_0$ and $\widehat{\mu}_1$ for predicting outcomes. Shared layers project $\widehat{f}_0$ and $\widehat{f}_1$ into a common space.

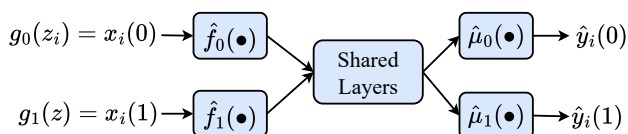

Figure 5: *SimPONet*'s model architecture.

## A.4 *SimPONet* Pseudocode

Here, we present the *SimPONet* pseudocode. The steps involved in our algorithm are:

line 1  First we use the simulator dataset $D_{\text{syn}}$ to apply contrastive losses on the counterfactual covariates using Eq. 1. This optimization gives us a $Z$ extractor in the simulator distribution, which we denote as $\widetilde{f}_t^S$.

line 2  We partition the training dataset into train, validation dataset using stratified split based on $T$. We then initialize the loss weigts $\lambda_f, \lambda_\tau$ to their defaults.

lines 4,5  We now decide upon the loss weight $\lambda_f$. To do so, we train RealOnly and $\text{Real}_\mu\text{Sim}_f$ models. We then assess the factual prediction errors of these models on the validation split of the training dataset. If RealOnly model performs much better than the $\text{Real}_\mu\text{Sim}_f$ model, it means that the $\widetilde{f}_t^S$ obtained from line2 above is of inferior quality. Therefore, we scale $\lambda_f$ that regularizes the *SimPONet*'s $\widehat{f}_t$ based on $\widetilde{f}_t^S$ to a very small value, 1e-4.

line 6  We can now apply gradient descent algorithm on the *SimPONet*'s objective in Eq. 3 to train the $\widehat{\mu}_t, \widehat{f}_t$ parameters of the model.

---

**Algorithm 1** *SimPONet* Algorithm

---

**Require:** Observational Data $D_{\mathrm{trn}}$: $\{(\boldsymbol{x}_i,t_i,y_i)\}$, Simulator Data $D_{\mathrm{syn}}$: $\{(\boldsymbol{x}_i^S(0),\boldsymbol{x}_i^S(1),y_i^S(0),y_i^S(1))\}$
 1: Let $\{\widehat{f}(\bullet,t)\}\leftarrow Z$ extraction functions, and $\{\widehat{\mu}(\bullet,t)\}\leftarrow$ outcome functions for $t=0,1$.
 2: Let $\widetilde{f}_t^S\leftarrow$ Eq. 1 (Minimize Contrastive loss on $D_{\mathrm{syn}}$)
 3: Set $D_{\mathrm{trn}},D_{\mathrm{val}}\leftarrow$ SPLIT($D_{\mathrm{trn}},pc=0.3$, stratify$=T$), and init default hyperparameters $\lambda_f,\lambda_\tau\leftarrow 1,1$
 4: RealOnly$\leftarrow\min_{\{\widehat{\mu},\widehat{f}\}}\sum_{D_{\mathrm{trn}}}(y_i-\widehat{\mu}_{t_i}(\widehat{f}_{t_i}(\boldsymbol{x}_i)))^2$; Real$_\mu$Sim$_f\leftarrow\min_{\widehat{\mu}}\sum_{D_{\mathrm{trn}}}(y_i-\widehat{\mu}_{t_i}(\widetilde{f}_{t_i}^S(\boldsymbol{x}_i)))^2$
 5: Set $\lambda_f\leftarrow$ 1e-4 if `FactualErr`(RealOnly$,D_{\mathrm{val}}$)$>>$`FactualErr`(Real$_\mu$Sim$_f,D_{\mathrm{val}}$)
 6: $\{\widehat{f}_t,\widehat{\mu}_t\}\leftarrow$ Eq. 3 (perform gradient descent on *SimPONet*'s objective using $D_{\mathrm{trn}},D_{\mathrm{syn}}$ while early stopping using Factual Error on $D_{\mathrm{val}}$)
 7: **Return** $\{\widehat{f}_t,\widehat{\mu}_t\}$ for $t=0,1$

---

We present the pseudocode for *SimPONet* in Alg. 1.

### A.5 Theoretical Analysis

In this section, we present the proofs for our theoretical results.

#### A.5.1 Proof of Lemma 2

The CATE error is related to the factual and counterfactual error as: $\mathcal{E}_{\mathrm{CATE}}^t\leq 2\mathcal{E}_F^t+2\mathcal{E}_{CF}^t$

**Proof.** We decompose the CATE error into factual and counterfactual estimation error as follows:

$$\mathcal{E}_{\mathrm{CATE}}^t=\int_{\boldsymbol{x}\in\mathcal{X}}[\tau_X(\boldsymbol{x},t)-\widehat{\tau_X}(\boldsymbol{x},t)]^2P(\boldsymbol{x}|t)d\boldsymbol{x}=\int_{\boldsymbol{x}\in\mathcal{X}}[\tau(f_t(\boldsymbol{x}))-\widehat{\tau}(\widehat{f}_t(\boldsymbol{x}))]^2P(\boldsymbol{x}|t)d\boldsymbol{x}$$
$$=\int_{\boldsymbol{x}\in\mathcal{X}}[(\mu_1(f_t(\boldsymbol{x}))-\mu_0(f_t(\boldsymbol{x})))-(\widehat{\mu}_1(\widehat{f}_t(\boldsymbol{x}))-\widehat{\mu}_0(\widehat{f}_t(\boldsymbol{x})))]^2P(\boldsymbol{x}|t)d\boldsymbol{x}$$
$$=\int_{\boldsymbol{x}\in\mathcal{X}}[(\mu_1(f_t(\boldsymbol{x}))-\widehat{\mu}_1(\widehat{f}_t(\boldsymbol{x})))-(\mu_0(f_t(\boldsymbol{x}))-\widehat{\mu}_0(\widehat{f}_t(\boldsymbol{x})))]^2P(\boldsymbol{x}|t)d\boldsymbol{x}$$

Let $t'=1-t$ denote the *counterfactual* treatment. We can then rewrite the above expression as:

$$\mathcal{E}_{\mathrm{CATE}}^t=\int_{\boldsymbol{x}\in\mathcal{X}}[(\mu_t(f_t(\boldsymbol{x}))-\widehat{\mu}_t(\widehat{f}_t(\boldsymbol{x})))-(\mu_{t'}(f_t(\boldsymbol{x}))-\widehat{\mu}_{t'}(\widehat{f}_t(\boldsymbol{x})))]^2P(\boldsymbol{x}|t)d\boldsymbol{x}$$

Now using the inequality $(a-b)^2\leq 2a^2+2b^2$, we can separate the *factual* and *counterfactual* terms:

$$\mathcal{E}_{\mathrm{CATE}}^t\leq 2\int_{\boldsymbol{x}\in\mathcal{X}}[\mu_t(f_t(\boldsymbol{x}))-\widehat{\mu}_t(\widehat{f}_t(\boldsymbol{x}))]^2P(\boldsymbol{x}|t)d\boldsymbol{x}+2\int_{\boldsymbol{x}\in\mathcal{X}}[\mu_{t'}(f_t(\boldsymbol{x}))-\widehat{\mu}_{t'}(\widehat{f}_t(\boldsymbol{x}))]^2P(\boldsymbol{x}|t)d\boldsymbol{x}$$
$$=2\mathcal{E}_F^t+2\mathcal{E}_{CF}^t$$

#### A.5.2 Recovery of $f^S$ upto a diffeomorphic transformation

**Lemma 5.** *As $|D_{syn}|\to\infty$, contrastive training with paired covariates recovers $\widetilde{f}_t^S=h\circ f_t^S$ while paired outcome supervision recovers $\widetilde{\tau}^S=\tau^S\circ h^{-1}$ where $h$ is a diffeomorphic transformation. Moreover, when the latent space $\mathcal{Z}\subset\mathbb{S}^{(n_z-1)}$ (unit-norm hypersphere in $\mathbb{R}^{n_z}$), $h$ is a rotation transform by Extended Mazur-Ulam Theorem as shown in (Zimmermann et al., 2021) (Proposition 2).*

**Proof.** Theorem 4.4 of (Von Kügelgen et al., 2021) shows that contrastive training with covariate pairs $\{\boldsymbol{x}_i^S(0),\boldsymbol{x}_i^S(1)\}$ recovers $Z$ upto a diffeomorphic transformation $h$, i.e. for the simulator DGP our estimate $\widehat{z}_i=\widetilde{f}^S(\boldsymbol{x}_i^S(t),t)=h(z_i)=h(f^S(\boldsymbol{x}_i^S(t),t),\forall t\in\mathcal{T}$. Moreover for unit-norm latent representations, $\mathcal{Z}\subset$

$\mathbb{S}^{d_z-1}$, (Zimmermann et al., 2021) show that $h$ is an isometric (norm-preserving) function and therefore, a rotation transform by an extension of Mazur-Ulam Theorem. Mazur-Ulam Theorem states that any smooth, invertible and isometric function is necessarily affine. Moreover, in our setting, the norm of $z$ as well as $\hat{z}$ is always one and thus, $h$ is necessarily a rotation. Therefore, we recover $\widetilde{f}^S = h \circ f^S$ upto a rotation of the true inverse map $f^S$ with sufficient paired samples from the simulator.

Next, we recover $\widetilde{\tau}^S$ from the following minimisation:

$$\widetilde{\tau}^S = \underset{\widehat{\tau}^S}{\mathrm{argmin}}\, \mathbb{E}_{\boldsymbol{x}^S}\left[\widehat{\tau}^S(\widetilde{f}^S(\boldsymbol{x}^S(t),t)) - \tau^S(f^S(\boldsymbol{x}^S(t),t))\right]^2 = \underset{\widehat{\tau}^S}{\mathrm{argmin}}\, \mathbb{E}_z\left[\widehat{\tau}^S(h(z)) - \tau^S(z)\right]^2$$

The above optimization gives $\widetilde{\tau}^S = \tau^S \circ h^{-1}$ and hence we recover the CATE function $\tau^S$ for the simulator DGP composed with $h^{-1}$.

**Proof of Lemma 4.**

Assume $\tau$ is $K_\tau$-Lipschitz, and $\widetilde{f}^S$ and $\widetilde{\tau}^S$ are estimates from the simulator DGP obtained from the optimization in Eq. 1, 2. Then, the CATE error on the estimates $\widehat{f}_t$ and $\widehat{\tau}$ admits the following bound:

$$\mathcal{E}^t_{\mathrm{CATE}}(\widehat{f}_t,\widehat{\tau}) \le [8\mathcal{E}^t_F + 12d_h(\widehat{\tau},\widetilde{\tau}^S) + 12K_\tau^2 d_{\boldsymbol{x}|t}(\widehat{f}_t,\widetilde{f}^S_t)] + [12d_z(\tau,\tau^S) + 12K_\tau^2 d_{\boldsymbol{x}|t}(f_t,f^S_t)]$$

where $d_{\boldsymbol{x}|t}, d_z, d_{h(z)}$ are distance functions in Sec. 3 and $\mathcal{E}^t_F$ is the factual loss.

**Proof.** We now construct at upper bound on *counterfactual* error $\mathcal{E}^t_{CF}$ that relies on both observational data and simulator estimates to motivate the *SimPONet* objective:

$$\mathcal{E}^t_{CF} = \int_{\boldsymbol{x}\in\mathcal{X}} [\mu_{t'}(f_t(\boldsymbol{x})) - \widehat{\mu}_{t'}(\widehat{f}_t(\boldsymbol{x}))]^2 P(\boldsymbol{x}|t)d\boldsymbol{x}$$

$$= \int_{\boldsymbol{x}\in\mathcal{X}} [(\mu_{t'}(f_t(\boldsymbol{x})) - \mu_t(f_t(\boldsymbol{x}))) - (\widehat{\mu}_{t'}(\widehat{f}_t(\boldsymbol{x})) - \widehat{\mu}_t(\widehat{f}_t(\boldsymbol{x}))) + \mu_t(f_t(\boldsymbol{x})) - \widehat{\mu}_t(\widehat{f}_t(\boldsymbol{x}))]^2 P(\boldsymbol{x}|t)d\boldsymbol{x}$$

$$= \int_{\boldsymbol{x}\in\mathcal{X}} [(2\mathbf{1}_{t=0}-1)\cdot(\tau(f_t(\boldsymbol{x})) - \widehat{\tau}(\widehat{f}_t(\boldsymbol{x}))) + \mu_t(f_t(\boldsymbol{x})) - \widehat{\mu}_t(\widehat{f}_t(\boldsymbol{x}))]^2 P(\boldsymbol{x}|t)d\boldsymbol{x}$$

Where $\mathbf{1}_{t=0} = 1$ when $t=0$ and zero otherwise, and thus, $(2\mathbf{1}_{t=0}-1) = \pm 1$ adjusting the sign of CATE terms. Now we utilise the inequality $(a+b+c)^2 \le 3(a^2+b^2+c^2)$ to obtain:

$$\mathcal{E}^t_{CF} = \int_{\boldsymbol{x}\in\mathcal{X}} [(2\mathbf{1}_{t=0}-1)\cdot(\tau(f_t(\boldsymbol{x})) - \widehat{\tau}(h\circ f_t(\boldsymbol{x})) + \widehat{\tau}(h\circ f_t(\boldsymbol{x})) - \widehat{\tau}(\widehat{f}_t(\boldsymbol{x}))) + \mu_t(f_t(\boldsymbol{x})) - \widehat{\mu}_t(\widehat{f}_t(\boldsymbol{x}))]^2 P(\boldsymbol{x}|t)d\boldsymbol{x}$$

$$\le 3\int_{\boldsymbol{x}\in\mathcal{X}} [\tau(f_t(\boldsymbol{x})) - \widehat{\tau}(h\circ f_t(\boldsymbol{x}))]^2 P(\boldsymbol{x}|t)d\boldsymbol{x} + 3\int_{\boldsymbol{x}\in\mathcal{X}} [\widehat{\tau}(h\circ f_t(\boldsymbol{x})) - \widehat{\tau}(\widehat{f}_t(\boldsymbol{x}))]^2 P(\boldsymbol{x}|t)d\boldsymbol{x}$$

$$+ 3\int_{\boldsymbol{x}\in\mathcal{X}} [\mu_t(f_t(\boldsymbol{x})) - \widehat{\mu}_t(\widehat{f}_t(\boldsymbol{x}))]^2 P(\boldsymbol{x}|t)d\boldsymbol{x}$$

$$= 3\int_{z\in\mathcal{Z}} [\tau(z) - \widehat{\tau}(h(z))]^2 P(z|t)dz + 3\int_{\boldsymbol{x}\in\mathcal{X}} [\widehat{\tau}(h(f_t(\boldsymbol{x}))) - \widehat{\tau}(\widehat{f}_t(\boldsymbol{x}))]^2 P(\boldsymbol{x}|t)d\boldsymbol{x} + 3\mathcal{E}^t_F$$

Here $h$ denotes the unknown rotation transformation that relates the estimated simulator functions $(\widetilde{f}^S, \widetilde{\tau}^S)$ with the ground-truth simulator functions $(f^S, \tau^S)$ as shown in Lemma 5. Let $K_\tau$ be the Lipschitz constant for $\widehat{\tau}$. We can bound the second term in the above expression as follows:

$$\mathcal{E}_{CF}^t \le 3\int_{z\in\mathcal{Z}}[\tau(z)-\widehat{\tau}(h(z))]^2 P(z|t)dz + 3K_\tau^2\int_{\boldsymbol{x}\in\mathcal{X}}||h(f_t(\boldsymbol{x}))-\widehat{f}_t(\boldsymbol{x})||^2 P(\boldsymbol{x}|t)d\boldsymbol{x} + 3\mathcal{E}_F^t$$
$$= 3d_z(\tau,\widehat{\tau}\circ h) + 3K_\tau^2 d_{\boldsymbol{x}|t}(h\circ f_t,\widehat{f}_t) + 3\mathcal{E}_F^t$$

Now we can add and subtract simulator function estimates to bound the two distance terms as follows:

$$\mathcal{E}_{CF}^t \le 3\int_{z\in\mathcal{Z}}[\tau(z)-\tau^S(z)+\tau^S(z)-\widehat{\tau}(h(z))]^2 P(z|t)dz$$
$$+ 3K_\tau^2\int_{\boldsymbol{x}\in\mathcal{X}}||h(f_t(\boldsymbol{x}))-h(f_t^S(\boldsymbol{x}))+h(f_t^S(\boldsymbol{x}))-\widehat{f}_t(\boldsymbol{x})||^2 P(\boldsymbol{x}|t)d\boldsymbol{x} + 3\mathcal{E}_F^t$$
$$\le 6\int_{z\in\mathcal{Z}}[\tau(z)-\tau^S(z)]^2 P(z|t)dz + 6\int_{z\in\mathcal{Z}}[\tau^S(z)-\widehat{\tau}(h(z))]^2 P(z|t)dz$$
$$+ 6K_\tau^2\int_{\boldsymbol{x}\in\mathcal{X}}||h(f_t(\boldsymbol{x}))-h(f_t^S(\boldsymbol{x}))||^2 P(\boldsymbol{x}|t)d\boldsymbol{x} + 6K_\tau^2\int_{\boldsymbol{x}\in\mathcal{X}}||h(f_t^S(\boldsymbol{x}))-\widehat{f}_t(\boldsymbol{x})||^2 P(\boldsymbol{x}|t)d\boldsymbol{x} + 3\mathcal{E}_F^t$$
$$= 6d_z(\tau,\tau^S) + 6d_z(\widehat{\tau}\circ h,\tau^S) + 6K_\tau^2 d_{\boldsymbol{x}|t}(h\circ f_t,h\circ f_t^S) + 6K_\tau^2 d_{\boldsymbol{x}|t}(\widehat{f}_t,h\circ f_t^S) + 3\mathcal{E}_F^t$$

Now, using Lemma 5, we can rewrite $\tau^S = \widetilde{\tau}^S\circ h$ in the second term. Thus, $d_z(\widehat{\tau}\circ h,\tau^S)=d_z(\widehat{\tau}\circ h,\widetilde{\tau}^S\circ h)$. Now making use of Definition 2, we can rewrite this as $d_{h(z)}(\widehat{\tau},\widetilde{\tau}^S)$ which is a distance function defined on the space of rotated latents $h(z)$. We also rewrite $h\circ f^S$ as $\widetilde{f}^S$ in the fourth term.

Moreover, $d_{\boldsymbol{x}|t}(h\circ f_t,h\circ f_t^S)=d_{\boldsymbol{x}|t}(f_t,f_t^S)$ since $h$ is a rotation transform and preserves the distance between any two vectors. Thus, $||f_t(\boldsymbol{x})-f_t^S(\boldsymbol{x})||_2 = ||h\circ f_t(\boldsymbol{x})-h\circ f_t^S(\boldsymbol{x})||_2$. Combining these results, we can evaluate the above bound to the following:

$$\mathcal{E}_{CF}^t \le [6d_{h(z)}(\widehat{\tau},\widetilde{\tau}^S) + 6K_\tau^2 d_{\boldsymbol{x}|t}(\widehat{f}_t,\widetilde{f}_t^S) + 3\mathcal{E}_F^t] + [6d_z(\tau,\tau^S) + 6K_\tau^2 d_{\boldsymbol{x}|t}(f_t,f_t^S)]$$

### A.6  Linear DGP Derivation

We derive expressions for CATE estimates $\widehat{\tau_X}(\boldsymbol{x},t)$ as well as $\mathcal{E}_{\text{CATE}}^t$ for each of our proposed estimators in the linear setting below. Note that ground truth CATE $\tau_X(\boldsymbol{x},t)=\boldsymbol{x}\boldsymbol{R}_t^{-1}(w_1-w_0)$. We consider factual treatment $t=1$ to illustrate the errors.

#### A.6.1  SimOnly

For SimOnly, we use $\widehat{\boldsymbol{R}}_t^{-1}=\boldsymbol{S}_t^{-1}$ and $\widehat{w}_t=w_t^S$ which are obtained by training on simulator data. Thus, the CATE estimate $\widehat{\tau_X}(\boldsymbol{x},t)=\boldsymbol{x}\boldsymbol{S}_t^{-1}(w_1^S-w_0^S)$. The CATE error on a sample $\boldsymbol{x}^*$, with treatment $t=1$ is given by $[\widehat{\tau_X}(\boldsymbol{x}^*,1)-\tau_X(\boldsymbol{x}^*,1)]^2 = [(\boldsymbol{x}^*(\boldsymbol{S}_1^{-1}(w_1^S-w_0^S)-\boldsymbol{R}_1^{-1}(w_1-w_0))]^2$

#### A.6.2  RealOnly

For RealOnly, the factual objective $\mathcal{E}_F^t = ||\boldsymbol{x}\widehat{\boldsymbol{R}}_t^{-1}\widehat{w}_t - y||_2^2 = ||\boldsymbol{x}\widehat{\boldsymbol{R}}_t^{-1}\widehat{w}_t - \boldsymbol{x}\boldsymbol{R}_t^{-1}w_t||_2^2$. Thus, the closed form solution of the estimator $\widehat{\boldsymbol{R}}_t^{-1}\widehat{w}_t=\boldsymbol{R}_t^{-1}w_t,\forall t\in\mathcal{T}$. Since we can't decouple the terms $\widehat{\boldsymbol{R}}_t^{-1}$ and $\widehat{w}_t$, the CATE estimate is given by $\widehat{\tau_X}(\boldsymbol{x}^*,t)=\boldsymbol{x}^*\widehat{\boldsymbol{R}}_1^{-1}\widehat{w}_1-\boldsymbol{x}^*\widehat{\boldsymbol{R}}_0^{-1}\widehat{w}_0=\boldsymbol{x}^*\boldsymbol{R}_1^{-1}w_1-\boldsymbol{x}^*\boldsymbol{R}_0^{-1}w_0$.
CATE error on sample $\boldsymbol{x}^*$ with treatment $t=1$ is given by $[\widehat{\tau_X}(\boldsymbol{x}^*,1)-\tau_X(\boldsymbol{x}^*,1)]^2 = [(\boldsymbol{x}^*\boldsymbol{R}_1^{-1}w_1-\boldsymbol{x}^*\boldsymbol{R}_0^{-1}w_0)-\boldsymbol{x}\boldsymbol{R}_1^{-1}(w_1-w_0)]^2 = [\boldsymbol{x}(\boldsymbol{R}_1^{-1}-\boldsymbol{R}_0^{-1})w_0]^2$

#### A.6.3  Real$_\mu$Sim$_f$

For Real$_\mu$Sim$_f$, we first set $\widehat{\boldsymbol{R}}_t^{-1}=\boldsymbol{S}_t^{-1}$ which is obtained by training on simulator data. Next, we train $\widehat{w}_t$ on the factual objective: $||\boldsymbol{x}\widehat{\boldsymbol{R}}_t^{-1}\widehat{w}_t-\boldsymbol{x}\boldsymbol{R}_t^{-1}w_t||_2^2 = ||\boldsymbol{x}\boldsymbol{S}_t^{-1}\widehat{w}_t-\boldsymbol{x}\boldsymbol{R}_t^{-1}w_t||_2^2$. This, gives us a closed form solution for the

minimising $\hat{w}_t = S_t R_t^{-1} w_t$. The CATE estimate $\widehat{\tau_X}(x^*,t) = x^* S_t^{-1}(\hat{w}_1 - \hat{w}_0) = x^* S_t^{-1}(S_1 R_1^{-1} w_1 - S_0 R_0^{-1} w_0)$. Fixing treatment $t = 1$, this simplifies further: $\widehat{\tau_X}(x^*,1) = x^* S_1^{-1}(S_1 R_1^{-1} w_1 - S_0 R_0^{-1} w_0) = x^*(R_1^{-1} w_1 - S_1^{-1} S_0 R_0^{-1} w_0)$. CATE Error is given by $[\widehat{\tau_X}(x^*,1) - \tau_X(x^*,1)]^2 = [x^*(R_1^{-1} w_1 - S_1^{-1} S_0 R_0^{-1} w_0) - x^* R_1^{-1}(w_1 - w_0)]^2 = [x^* R_1^{-1} w_0 - x^* S_1^{-1} S_0 R_0^{-1} w_0]^2 = [x^*(R_1^{-1} - S_1^{-1} S_0 R_0^{-1}) w_0]^2$

Table 7: This table presents the predicted CATE and the corresponding CATE errors obtained from the three CATE proposals computed analytically for a test instance $x^\star$ observed under treatment 1.

| Method | Estimate for CATE $\widehat{\tau_X}(x^\star,1)$ | CATE Error $[\widehat{\tau_X}(x^\star,1) - \tau(x^\star,1)]^2$ |
|---|---|---|
| SimOnly | $x^\star S_1^{-1} w_\tau^S$ | $\left[x^\star\left(R_1^{-1} w_\tau - S_1^{-1} w_\tau^S\right)\right]^2$ |
| RealOnly | $x^\star\left(R_1^{-1} w_1 - R_0^{-1} w_0\right)$ | $\left[x^\star\left(R_0^{-1} - R_1^{-1}\right) w_0\right]^2$ |
| $\text{Real}_\mu \text{Sim}_f$ | $x^\star S_1^{-1} S_1 R_1^{-1} w_1 - x^\star S_1^{-1} S_0 R_0^{-1} w_0$ | $\left[x^\star(R_1^{-1} - S_1^{-1} S_0 R_0^{-1}) w_0\right]^2$ |

### A.6.4 *SimPONet*

We train both $\hat{R}_t^{-1}, \hat{w}_t$ on the following objective jointly:

$$\mathcal{L}(\{\hat{R}_t^{-1}, \hat{w}_t\}_{t=0,1}) = \left[\sum_{t=0,1} ||x\hat{R}_t^{-1}\hat{w}_t - x R_t^{-1} w_t||_2^2 + \lambda_f \sum_{t=0,1} ||x\hat{R}_t^{-1} - x S_t^{-1}||_F^2 + \lambda_\tau ||\mathbf{z}(\hat{w}_1 - \hat{w}_0) - \mathbf{z}(w_1 - w_0)||_2^2\right]$$

Here, $\mathbf{z} = x_{t'}^S S_{t'}^{-1}$ are the latents for simulated covariates $x_{t'}^S$ (which are identifiable from $D_\text{syn}$). Due to the joint nature of this optimisation, it is not possible to derive closed form solutions for the optimum. However, one can compuet gradients of the objective with respect to $\hat{R}_t^{-1}$ and $\hat{w}_t$ separately. This, gives us an alternating minimisation algorithm with closed form updates.

$$\frac{\partial \mathcal{L}}{\partial \hat{R}_t^{-1}} = \frac{\partial}{\partial \hat{R}_t^{-1}}\left[||x\hat{R}_t^{-1}\hat{w}_t - y||_2^2 + \lambda_f ||x\hat{R}_t^{-1} - x S_t^{-1}||_F^2\right]$$
$$= 2x^T x\hat{R}_t^{-1}(\hat{w}_t \hat{w}_t^T + \lambda_f I) - 2x^T y\hat{w}_t + -2\lambda_f x^T x S_t^{-1}$$

Setting the derivative to zero, we obtain the following update rule:

$$\hat{R}_t^{-1} \leftarrow (x^\dagger y\hat{w}_t + \lambda_f S_t^{-1}) \cdot (\hat{w}_t \hat{w}_t^T + \lambda_f I)^{-1}$$

where $x^\dagger = (x^T x)^{-1} x^T$ is the pseudoinverse of $x$.

$$\frac{\partial \mathcal{L}}{\partial \hat{w}_t} = \frac{\partial}{\partial \hat{w}_t}\left[||x\hat{R}_t^{-1}\hat{w}_t - y||_2^2 + \lambda_\tau ||\mathbf{z}(\hat{w}_t - \hat{w}_{t'}) - (y_1^S - y_0^S)||_2^2\right]$$
$$= 2(\hat{z}^T \hat{z})\hat{w}_t - 2\hat{z}^T y + 2\lambda_\tau(z^T z\hat{w}_t - z^T(z\hat{w}_{t'} + (y_1^S - y_0^S)))$$
$$= 2[(\hat{z}^T \hat{z}) + \lambda_\tau(z^T z)]\hat{w}_t - 2(\hat{z}^T y + \lambda_\tau z^T(z\hat{w}_{t'} + (y_1^S - y_0^S)))$$

Where $\hat{z} = x\hat{R}_t^{-1}$. Setting the derivative to zero, we obtain the following update rule:

$$\hat{w}_t \leftarrow ((\hat{z}^T \hat{z}) + \lambda_\tau(z^T z))^{-1} \cdot (\hat{z}^T y + \lambda_\tau z^T(z\hat{w}_{t'} + (y_1^S - y_0^S)))$$

For *SimPONet*, we perform alternating updates of $\hat{w}_t$ and $\hat{R}_t^{-1}$ fixing the other estimate.

### A.7 Summary of Datasets

**IHDP.** The Infant Health and Development Program (IHDP) is a randomized controlled trial designed to assess the impact of physician home visits on the cognitive test performance of premature infants. The dataset exhibits selection bias due to the deliberate removal of non-random subsets of treated individuals from the training data. Since outcomes are observed for only one treatment, we generate both observed and counterfactual outcomes using a synthetic outcome generation function based on the original covariates for both treatments, making the dataset suitable for causal inference.

Table 8: Effect of varying training sizes on CATE using the IHDP dataset. We experiment with training proportions of 10%, 25%, 50%, and 75% of the full training set. Results demonstrate how the performance of *SimPONet* and baseline methods evolves with varying amounts of training data.

| Training Percentage | 0.10 | 0.25 | 0.50 | 0.75 |
|---|---|---|---|---|
| RNet | 3.08 (0.08) | 2.52 (0.07) | 2.30 (0.12) | 2.30 (0.14) |
| XNet | 2.43 (0.09) | 1.90 (0.12) | 1.16 (0.19) | 1.10 (0.35) |
| DRNet | 3.28 (0.04) | 2.01 (0.03) | 1.05 (0.40) | 1.05 (0.44) |
| CFRNet | 1.41 (0.03) | 1.06 (0.21) | 0.95 (0.59) | 1.02 (0.50) |
| FlexTENet | 2.97 (0.08) | 2.33 (0.07) | 1.04 (0.40) | 1.03 (0.49) |
| DragonNet | 3.48 (0.04) | 2.02 (0.04) | 0.97 (0.53) | 1.02 (0.50) |
| IPW | 3.44 (0.03) | 1.74 (0.06) | **0.94 (0.60)** | 1.03 (0.49) |
| NearNeighbor | 2.10 (0.19) | 1.65 (0.05) | 2.65 (0.11) | 1.07 (0.47) |
| PerfectMatch | 3.44 (0.03) | 2.54 (0.03) | 3.32 (0.01) | 1.05 (0.45) |
| PairNet | 1.64 (0.13) | 1.18 (0.11) | 1.09 (0.30) | 1.09 (0.36) |
| SimOnly | 1.45 (0.23) | 1.45 (0.07) | 1.45 (0.15) | 1.45 (0.04) |
| RealOnly | 2.28 (0.13) | 2.92 (0.07) | 0.97 (0.54) | **0.86 (0.75)** |
| $Real_\mu Sim_f$ | 3.35 (0.03) | 2.01 (0.04) | 1.07 (0.31) | 1.02 (0.25) |
| *SimPONet* | **1.01 (0.00)** | **0.93 (0.00)** | 0.95 (0.00) | 0.87 (0.00) |

The IHDP dataset includes 747 subjects and 25 variables. While the original dataset discussed in (Shalit et al., 2017) had 1000 versions, our work uses a smaller version with 100 iterations, aligning with the CATENets benchmark. Each version varies in the complexity of the assumed outcome generation function, treatment effect heterogeneity, etc. As outlined in (Curth et al., 2021), reporting the standard deviation of performance across the 100 different seeds is inappropriate. Therefore, we calculate $p$-values through paired t-tests between our method (*SimPONet*) and other baseline methods, using *SimPONet* as the baseline for all experiments. We follow setting D of the IHDP dataset as mentioned in (Curth & van der Schaar, 2021) where response surfaces are modified to suppress the extremely high variance of potential outcomes in certain versions of the IHDP dataset.

**ACIC.** The Atlantic Causal Inference Conference (ACIC) competition dataset $(2016)^4$ consists of 77 datasets, all containing the same 58 covariates derived from the Collaborative Perinatal Project. Each dataset simulates binary treatment assignments and continuous outcome variables, with variations in the complexity of the treatment assignment mechanism, treatment effect heterogeneity, the ratio of treated to control observations, overlap between treatment and control groups, dimensionality of the confounder space, and the magnitude of the treatment effect.

All datasets share common characteristics, such as independent and identically distributed observations conditional on covariates, adherence to the ignorability assumption (selection on observables with all confounders measured and no hidden bias), and the presence of non-true confounding covariates. Of the 77 datasets, we selected a subset of three: versions 2, 7, and 26, aligning with the CATENets benchmark. These versions present non-linear covariate-to-outcome relationships and maximum variability in treatment effect heterogeneity. Version 2, notably, exhibits no heterogeneity, meaning the treatment effect is constant across all individuals. However, accurately estimating outcome differences even for this version is challenging due to the inherent noise in potential outcome realizations in the dataset.

### A.8 Experiments with Limited Training Data

In this experiment, we evaluate the performance of CATE methods with varying training sizes. We use 10 randomly selected dataset versions of the IHDP dataset. For each version, the methods are trained using 10%, 25%, 50%, and 75% of the training data. Importantly, the simulator dataset is not subsampled, so the SimOnly model maintains the same CATE error across all training sizes. The results are presented in Table 8. At

---

[4]https://jenniferhill7.wixsite.com/acic-2016/competition

extremely low training sizes (10%), *SimPONet* achieves the lowest CATE error with a significant margin over all baselines. With 25% of the training data, the baseline methods improve, but *SimPONet* continues to deliver the best CATE error. At 50% training size, the baselines further improve and perform comparably to *SimPONet*. While *SimPONet*'s CATE error increases slightly in this setting, it remains close, trailing the best-performing baseline by only 0.01. Finally, at 75% training size, *SimPONet*'s CATE error decreases significantly, while many baselines plateau in performance. Overall, *SimPONet* demonstrates exceptional robustness in estimating CATE, particularly in limited training data scenarios.

## A.9 Table of Symbols

| Symbol | Definition |
|---|---|
| $X$ | Real post-treatment covariates: Random Variable |
| $Y$ | Real outcomes: Random Variable |
| $X^S$ | Simulator post-treatment covariates: Random Variable |
| $Y^S$ | Simulator outcomes: Random Variable |
| $T$ | Treatment: Random Variable |
| $Z$ | Latent (unobserved) pre-treatment representations: Random Variable |
| $D_{\mathrm{trn}}$ | Observational training dataset from Real DGP |
| $D_{\mathrm{syn}}$ | Counterfactual dataset from Simulator DGP |
| $D_{\mathrm{tst}}$ | Test dataset from Real DGP |
| $\boldsymbol{x},\boldsymbol{x}^S,z,t,y,y^S$ | Realisations of random variables $X,X^S,Z,T,Y,Y^S$ respectively |
| $\mathcal{X}$ | Space of post-treatment covariate values: Set |
| $\mathcal{T}$ | Space of treatment values: Set $=\{0,1\}$ |
| $\mathcal{Z}$ | Space of latents: Set |
| $\mathcal{Y}$ | Space of outcomes: Set |
| $n_z,n_x$ | Dimensions of vector spaces in which $\mathcal{Z},\mathcal{X}$ lie |
| $Y_i(t)$ | Potential outcome for $i^{\mathrm{th}}$ unit under treatment $t$ |
| $X_i(t)$ | Potential post-treatment covariate for $i^{\mathrm{th}}$ unit under treatment $t$ |
| $g_t$ | Mapping from $\mathcal{Z}\mapsto\mathcal{X}$, transforms latents to real post-treatment covariates under $t$ |
| $g_t^S$ | Mapping from $\mathcal{Z}\mapsto\mathcal{X}$, transforms latents to simulated post-treatment covariates under $t$ |
| $f_t$ | Mapping from $\mathcal{X}\mapsto\mathcal{Z}$, transforms real post-treatment covariates under $t$ to latents |
| $f_t^S$ | Mapping $\mathcal{X}\mapsto\mathcal{Z}$, transforms simulated post-treatment covariates under $t$ to latents |
| $P_Z$ | Probability distribution of latents $Z$ |
| $\mu_t$ | Outcome function for real data under $t$ |
| $\mu_t^S$ | Outcome function for simulated data under $t$ |
| $\tau$ | Conditional Average Treatment Effect for real data, $\mu_1-\mu_0$, Mapping $\mathcal{Z}\mapsto\mathcal{Y}$ |
| $\tau^S$ | Conditional Average Treatment Effect for simulated data, $\mu_1^S-\mu_0^S$, Mapping $\mathcal{Z}\mapsto\mathcal{Y}$ |
| $\circ$ | Composition of functions |
| $\tau_X(\boldsymbol{x},t)$ | Conditional Average Treatment Effect for real data, $\tau\circ f_t(\boldsymbol{x})$, Mapping $\mathcal{X}\times\mathcal{T}\mapsto\mathcal{Y}$ |
| $\tau_X^S(\boldsymbol{x}^S,t)$ | Conditional Average Treatment Effect for simulated data, $\tau^S\circ f_t^S(\boldsymbol{x}^S)$, Mapping $\mathcal{X}\times\mathcal{T}\mapsto\mathcal{Y}$ |
| $h$ | Diffeomorphic transformation, arises due to contrastive learning |
| $\mathbb{S}^d$ | Unit-norm hypersphere of dimension $d$, Subset of $\mathbb{R}^{(d+1)}$ |
| $d_{\boldsymbol{x}\mid t}$ | Expected squared-distance between two functions on $P(X\mid T)$, see Section 3 for definition |
| $d_z$ | Expected squared-distance between two functions on $P_Z$, see Section 3 for definition |
| $d_{h(z)}$ | $d_z$ under transformation $h$ on $z$, see Section 3 for definition |
| $\mathrm{sim}(\bullet,\bullet)$ | Cosine similarity |
| $\widehat{f_t}$ | Estimate for $f_t$ |
| $\widehat{f}_t^S$ | Estimate for $f_t^S$ |
| $\widehat{\mu}_t$ | Estimate for $\mu_t$ |
| $\widehat{\mu}_t^S$ | Estimate for $\mu_t^S$ |
| $\widetilde{f}_t^S$ | Estimate for $f_t^S$ recovered from contrastive learning |
| $\widetilde{\mu}_t^S$ | Estimate for $\mu_t^S$ on recovering Simulator DGP |
| $\mathcal{E}_{\mathrm{CATE}}$ | CATE estimation error |
| $\mathcal{E}_{\mathrm{CATE}}^t$ | CATE estimation error on covariates $\boldsymbol{x}$ under treatment $t$ |
| $\mathcal{E}_F^t$ | Factual error on treatment $t$ samples |
| $\mathcal{E}_{CF}^t$ | Counterfactual error on treatment $t$ samples |
| $K_\mu$ | Lipschitz constant for $\mu_t,\widehat{\mu}_t$ |
| $K_\tau$ | Lipschitz constant for $\tau,\widehat{\tau}$ |
| $K_{\mu^S}$ | Lipschitz constant for $\mu_t^S,\widehat{\mu}_t^S,\widetilde{\mu}_t^S$ |
| $K_{\tau^S}$ | Lipschitz constant for $\tau^S,\widehat{\tau}^S,\widetilde{\tau}^S$ |
| $\boldsymbol{R}_t$ | $g_t$ for linear DGP: Matrix |
| $\boldsymbol{S}_t$ | $g_t^S$ for linear DGP: Matrix |
| $w_t$ | $\mu_t$ for linear DGP: Vector |
| $w_t^S$ | $\mu_t^S$ for linear DGP: Vector |
| $w_\tau$ | $\tau$ for linear DGP: Vector |
| $w_\tau^S$ | $\tau^S$ for linear DGP: Vector |

