# OpenReview forum: "Leveraging a Simulator for Learning Causal Representations from Post-Treatment Covariates for CATE"
_TMLR — Accepted by TMLR_

### Review · Reviewer_q6UZ · 2024-11-08

**Summary Of Contributions:**

The paper focuses on estimating treatment effects based on covariates that are collected after treatment. This setting can appear in reality and brings some unique challenges, but has not been analyzed in prior work on ML for CATE estimation.

The authors analyze this problem. Starting with a general impossibility result, they describe the necessary assumptions for solving it. Different general strategies are compared. The insights from this comparison lead to a novel method, SimPONet. The strategies and method is analyzed empirically and compared to other benchmarks.

**Audience:**

Yes

**Broader Impact Concerns:**

/

**Claims And Evidence:**

No

**Requested Changes:**

## Presentation
- Some more practical, real-world examples of the considered setting would be appreciated. I feel like this would make the importance of the proposed work more clear.
- The work makes several assumptions, but does not discuss the feasibility of these assumptions in reality. Some examples:
  - $g_t$ being diffeomorphic is a strong assumption, which I could imagine not always being reasonable in practice. A discussion of some real cases where this assumption is violated would be useful. Additionally, an experiment analyzing your method under violations of this assumption would be insightful.
  - Why do you assume a dirac-delta and normal distribution for the DGP (Section 3)? These assumptions seem arbitrary and are not discussed.

## Tuning
- In my opinion, the authors do not give practical advice for tuning the loss weights $\lambda_f$ and $\lambda_g$ in Section 4. The default values are chosen in what seems to be an arbitrary manner. At the very least, more experiments need to be performed to judge the sensitivity of your method towards these hyperparameters.
- How were the neural network architectures and training procedures tuned (for your method and the benchmarks)?

## Experimental results
- Sensitivity analyses for the loss weights as described above.
- Why are three iterations of ACIC selected, instead of running the methods over all iterations, as is standard practice? I would understand differentiating between constant CATE and heterogeneous effects for ACIC, but that does not exclude any iterations.

___
### Minor points
- Abstract: "More realistic scenario": I think both collecting covariates pre- and post-treatment are realistic. There is no need to argue that one is more realistic than the other.
- p. 19: "Please refer [to] (Pearl, 2015) ..."

**Strengths And Weaknesses:**

## Strengths

- The work is presented clearly, and it is easy to follow.
- Clear description of the problem that is tackled, the challenges it presents.
- I like how the authors start from naive strategies and show why these can be suboptimal, to then use these insights to motivate the proposed method.
- A combination of theoretical and empirical results aiming to validate the method.

## Weaknesses

- The assumptions underlying the methodology need to be explained more clearly.
- It is not clear ow the method and benchmarks were tuned (or should be tuned).
- The experimental analysis needs to be improved.

---

> ### Author Response · Authors · 2024-11-27
> **Rebuttal Response**
>
> Thanks so much for appreciating our work and providing constructive feedback. Please see our responses below:
>
> > **Q1 Examples of real-world simulators**
>
> We have added two examples in Sec. 2.3. These two examples demonstrate the practical utility of simulators in assessing CATE: one in electrochemistry domain, and the other in pharmacology. We kindly request the reviewer to take a look.
>
> > **Q2 g_t is diffeomorphic**
>
> We kindly refer the reviewer to `Q3` from Reviewer `D6nV` for a detailed response. Additionally, we have included experiments in Section 5.6, where we chose $g_t$ to be an MLP that is not guaranteed to be invertible unlike the normalizing flows considered in our main results.
>
> > **Q3 dirac-delta**
>
> We have assumed the Dirac delta function to ensure invertibility. This assumption was essential for our theoretical analysis. In practice, the SimPONet algorithm can run even without this assumption as explained in the previous responses.
>
> > **Q4 Tune loss weights $\lambda_\tau, \lambda_f$**
>
> We analyze the sensitivity to the loss weights $\lambda_f$ and $\lambda_\tau$ using the IHDP covariates. We conduct this analysis across the three variants used in our study. For the Synthetic datasets, we set the real simulator gaps to high. In the Semi-Synthetic IHDP setting, we evaluate performance on five randomly selected dataset versions. We summarize the results in Table 6.
>
> Overall, we observe that SimPONet is sensitive to the choice of hyperparameter settings in non-linear and semi-synthetic versions, and remains fairly robust in the linear setting. However, tuning these hyperparameters is challenging in practice due to the absence of counterfactual data in real observational datasets. Despite this sensitivity, \our\ outperforms achieves either comparable performance or manages to surpass the baseline methods across different loss weight configurations.
>
>
>
> > **Q5 Neural architectures and hyper-parameters tuning**
>
> The neural architectures were identical across SimPONet and all the baselines. Since the baselines are directly fed the simulated causal representations, $ \tilde{f_s}(X) $, we employed three shared layers, each with 200 neurons, to project them into a common space. Additionally, two treatment-specific layers, each with 200 neurons, were used to predict the outcomes $ \mu_t $.
>
> Certain baselines like DragonNet, etc. that depends on propensity scores have additional layers with 200 neurons after the shared layers to predict $P(T|Z)$.
>
> For SimPONet, which involves learning the causal representations from $ X $, we included treatment-specific causal representation extractors, implemented as two-layered MLPs, each with 200 neurons. A pictorial representation of this architecture can be found in Fig. 5 of our appendix.
>
> All methods, including SimPONet, were trained for a maximum of 1000 epochs, with early stopping based on the factual validation error. We ensured consistency across all methods by setting appropriate random seeds so that each method used the same split for the training and validation datasets. During training we observed that almost all methods converged well before the 1000 epochs.
>
>
>
> > **Q6 All ACIC seeds**
>
> **Why we chose these 3 seeds**
>
> Out of the 77 dataset seeds, we selected three specific versions: 2, 7, and 26, in alignment with the CATENets benchmark (Curth et al.). These particular versions showcase non-linear covariate-to-outcome relationships and exhibit the highest variability in treatment effect heterogeneity among all 77 ACIC seeds. Version 2, in particular, demonstrates no heterogeneity, meaning the treatment effect remains constant across all individuals. The other selected versions (7 and 26) exhibit medium to large treatment effect heterogeneity. Previous works, including PairNet and Curth et al., have also focused on these seeds for a more fine-grained assessment.
>
> **All seeds**
>
> In addition, we have included results for all ACIC seeds under the column "ACIC-All" in Table 2. SimPONet continues to outperform all other methods in terms of the average CATE error. However, when compared across all seeds, some baselines yield $p$-values close to 0.4, indicating that, on certain dataset seeds, these baselines achieve results that are quite comparable to SimPONet.

---

> > ### Comment · Reviewer_q6UZ · 2024-12-16
> >
> > Thank you for your response and for your updates to the paper. In my opinion, this paper is both interesting and technically sound. Therefore, I recommend acceptance.

---

### Review · Reviewer_D6nV · 2024-11-10

**Summary Of Contributions:**

The paper offers an intriguing new perspective on CATE estimation by addressing a previously overlooked challenge. While most prior work assumes access to pre-treatment covariates to model CATE, this paper focuses on the scenario where only post-treatment covariates are available, rendering traditional assumptions invalid and CATE non-identifiable. By incorporating an off-the-shelf simulator, the authors demonstrate how it can be combined with a novel loss function to address this challenge, presenting a unique data-generating and learning process tailored to this setting.

**Audience:**

Yes

**Claims And Evidence:**

No

**Requested Changes:**

I would like the authors to address my questions in the weakness sections.
Some minor suggestions:

$\newline$

1. Consider adding a footnote to your first table explaining the metric and the meaning of the brackets, as these details are currently only mentioned in the Neural Architecture and Hyperparameters section, which readers may have difficulty locating.
2. Another advantage of your method may be its robustness with small sample sizes. By using the simulator to augment the sample size, it can improve performance and stability. It would be insightful to see this effect demonstrated by reducing the training data size to observe how your method's advantage over others changes.
3. An interesting experiment would be to test how the model's performance is affected when using an entirely uninformative simulator, to see if it can remain robust. In practice, simulator quality isn’t always guaranteed, so understanding the model's behavior in worst-case scenarios could provide valuable insights.

**Strengths And Weaknesses:**

Strength:
1. The paper introduces a unique approach to post-treatment CATE estimation, highlighting limitations in existing literature and demonstrating how the proposed method addresses these challenges.

2. Clear, step-by-step construction and rigorous proof showing how the proposed loss bounds the true CATE error.

3. Intuitive demonstrations of the contribution of each regularization terms through data.

Weakness:
1. A potential overstatement of the scope of the paper. it said: $\textit{In this paper, we address a more realistic scenario where both covariates and outcomes are gathered after treatment}$. While post-treatment covariates are challenging, substantial numbers of non-randomized trials and studies where selection bias arises from pre-treatment covariates that influence treatment choices.

2.  Limited practical value: the method may improve CATE estimation on existing data compared to current approaches. However, in many real-world applications, we aim to use learned functions to guide treatment recommendations for patients or individuals. In these cases, predictions are typically based on pre-treatment covariates.

3. The paper assumes a diffeomorphism of $\( g(t) \)$, making CATE identifiable for pre-treatment methods by establishing a bijective mapping $\( Z \leftrightarrow X \)$. Many existing methods apply separate functions to treatment and control groups, allowing conditioning on $\( X \)$ to serve the same purpose as conditioning on $\( Z \)$. This assumption makes CATE identifiable within the paper’s data generation framework.

4. For generality, pre-treatment can be considered a special case of post-treatment by assuming $X$ remains unchanged regardless of the treatment given. It’s promising that your proposed loss can bound the CATE. However, many causal methods that learn a latent space also establish bounds. Could you provide an apples-to-apples comparison showing that your theoretical bound is tighter in certain cases? For example, you could compare it with a classic method, such as CFRnet, which you cited in your paper.

5. In many cases, the simulator seems overly ideal, as on page 10 where $\tau^S(z) = \tau(z) + (\sigma(\tau) \gamma_\tau \cdot z^T \omega_\tau^S)$, which is also unbiased. To make the simulation more realistic, consider using a VAE approach to learn the observed data distribution. Otherwise, it seems like the model is being fed additional information, naturally giving it an advantage over other methods.

6. On page 10, you mention that for other competing methods, which were not developed for post-treatment settings, you use a learned $f_t(x)$. However, this might create an unfair comparison since these methods are not designed for this purpose and may therefore perform suboptimally. Given that pre-treatment covariates are a special case of post-treatment covariates, could you compare your method with others using only pre-treatment covariates? While your method doesn’t need to outperform them, demonstrating its robustness would provide valuable insight.

---

> ### Author Response · Authors · 2024-11-27
> **Rebuttal Response**
>
> Thanks so much for appreciating our work and constructive feedback. Please see our responses below.
>
> > **Q1: More Realistic Scenario**
>
> We have removed this phrase. While we agree that both settings are practically relevant, determining which one is more prominent is subjective, and we have adjusted the writing to reflect this. However, our claim that post-treatment methods have been relatively less-researched remains valid and highlights a gap that requires further attention in research.
>
> > **Q2: Applications on Pre-T Covariates**
>
> We fully agree with the reviewer that if pre-treatment covariates are available, SimPONet should not be used to estimate CATE. In such cases, since identifiability holds without the need for extracting causal representations, there is no explicit need for a simulator. Prior CATE methods proposed for pre-treatment covariates should be adequate, and used for such settings. We have added this as a remark in Sec 4.2 to this effect.
>
> > **Q3 Justify the diffeomorphic assumption**
>
> We motivate this with an experiment.
>
> Suppose there exists a latent $Z$ which acts as a sufficient adjustment set for CATE. To estimate $ Z $ from a post-T  $ X $, we need to be able to recover such a $ Z $ from it. Diffeomorphism guarantees that $ Z $ is preserved in $ X $, without which CATE estimation becomes hopeless. Under diffeomorphism, there exists a bijection from post-treatment $ X $ to latent $ Z $. In our theoretical analysis, diffeomorphism allows us to derive insightful bounds on CATE error. Further, diffeomorphism was also used in prior work [2] to theoretically show that contrastive learning on simulated covariate pairs provably recovers the diffeomorphic simulator DGP up to a rotation.
>
> Even for the easier CATE estimation from pre-treatment covariates $ W $, certain strong assumptions like “no unobserved confounders” are inevitable. In the same spirit, our diffeomorphism assumption is only meant to enforce “no unrecoverable latents” required to identify $ Z $ from $ X $.
>
> We expand on the “no unrecoverable latents” notion below, explaining how diffeomorphism may be relaxed:
>
> In practice, there may be several latent factors $ Z $ that give rise to $ X $ but $ Z $ may depend only on a subset $ Z_s \subset Z$ of them. The SimPONet algorithm would continue to work as long as the subset $ Z_s $ is recovered from $ X $. This is a relaxation of the diffeomorphic setting, since several different $ Z $ may have the same underlying $ Z_s $, which can be recovered through a many-to-one transformation $ g(Z) $.
>
> We test the above point empirically as follows: similar to the linear experiments in Section 5.2, we sample 10-dimensional Gaussian covariates $ Z $. Next, we define $ X $ as a linear function of the first-$ K $ coordinates of $ Z $. The space of optima for $ Z $ encompasses several rank-$ K $ matrices. SimPONet would give good CATE estimates if it is able to recover any one of them. We experimented with $ K $, and we show the results below.
>
> | $ K $ | SimOnly | RealOnly | SimPONet |
> |--|--|---|---|
> | 9       | 2.704 ± 0.36 | 15.038 ± 7.80 | 14.629 ± 9.97 |
> | 7       | 2.238 ± 0.20 | 18.925 ± 9.54 | 13.973 ± 9.35 |
> | 5       | 2.047 ± 0.56 | 43.447 ± 57.94 | 14.238 ± 9.27 |
> | 3       | 2.372 ± 0.40 | 108.069 ± 180.21 | 14.554 ± 9.41 |
>
> Next, we consider another setting where all the necessary latent factors $ Z $ are not recoverable from $ X $, making CATE non-identifiable. We test this setting empirically by sampling (10+$ K $)-dimensional Gaussian covariates $ Z $ and define $ Y $ as a function of this $ Z $. The observed covariates $ X $ are a function of only the first 10 entries in $ Z $, making other $ Z $ entries non-recoverable from $ X $. We experimented with $ K $, obtaining the following results:
>
> | $ K $ | SimOnly | RealOnly  | SimPONet |
> |--|--|---|---|
> | 5  | 3.536 ± 0.75 | 54.881 ± 108.30 | 3.365 ± 0.67 |
> | 10 | 3.693 ± 0.79 | 22.145 ± 14.55  | 3.479 ± 0.65 |
> | 15 | 4.086 ± 0.87 | 37.131 ± 44.74 | 3.886 ± 0.73 |
> | 20 | 3.981 ± 0.54 | 30.007 ± 43.37  | 3.932 ± 0.52 |
>
> SimPONet outperforms other approaches even when diffeomorphism is relaxed in both the identifiable and non-identifiable settings.
>
> Hence, diffeomorphism ensures that latent confounders are preserved in the observed post-treatment covariates, enabling CATE identifiability. This assumption is standard in the disentangled representation learning community [1, 2]. While it supports theoretical analysis, it is not explicitly required by the SimPONet algorithm. Furthermore, we have included experiments that relax this assumption in Sec. 5.7 of our revised paper.
>
> *References*
>
> [1] Locatello et. al. Challenging common assumptions in the unsupervised learning of disentangled representations. *International Conference on Machine Learning*, PMLR, 2019.
>
> [2] Kügelgen et. al. Self-supervised learning with data augmentations provably isolates content from style. *Advances in Neural Information Processing Systems*, 2021.

---

> > ### Author Response · Authors · 2024-11-27
> > **Rebuttal Response**
> >
> > > **Q4 Pre-T is a special case of post-T**
> >
> > Thanks for this insightful question. Indeed, we can treat Pre-T as a special case of post-T, but **only** under the condition when the functions $g_0 = g_1$. This corresponds to the ideal case of the RealOnly model, where we showed that if this holds, then RealOnly can achieve zero CATE error in the limit of the training size. For such settings, the CATE methods developed for pre-T covariates in the literature are adequate. We have added this useful remark under the RealOnly model in Sec. 4.2. This response further answers the reviewer’s question Q2.
> >
> > > **Q4 b Provide an apples-to-apples comparison with a method like CFRNet.**
> >
> > An apples-to-apples comparison is tricky because CFRNet addresses a setting where CATE is identifiable and provides bounds to characterize the CF error. Our setting is more challenging because CATE is not identifiable. The only analogy we could draw is that CFRNet’s CATE error is bounded by the IPM $(P(X|T=0), P(X|T=1))$ term, which measures the confounding distance. In contrast, our analysis bounds SimPONet’s CATE error based on the distance between the real and simulator distributions.
> >
> >
> >
> > > **Q5 Simulator is overly ideal**
> >
> > We ensured that our simulated data was non-ideal. For an ideal simulator, the `SimOnly` model would consistently be the best CATE estimator, which is not the case across several tables. Specifically, we synthesized the simulated $\tau^S$ for semi-synthetic datasets using the equation:
> >
> > $$
> > \tau^S(z) = \tau(z) + (\sigma(\tau) \cdot \gamma_\tau \cdot z^\top w_\tau^S),
> > $$
> >
> > where $\gamma_\tau$ controls the noise level.
> >
> > We acknowledge that the synthetic $\tau^S$ in our experiments was unbiased. However, we have added an experiment in Sec. 5.6 exploring varying values of $\gamma_\tau$. When $\gamma_\tau = 0$, the simulator is ideal. However, for $\gamma_\tau = 1$, the noise in $\tau^S$ has the same standard deviation as $\tau$, making the signal $\tau$ and the noise component of equal scale.  We observe that for a simulator with such degraded $\tau^S$ supervision, even though $\tau^S$ remains an unbiased estimate of $\tau$, the finite sample training causes a significant increase in the CATE error for the `SimOnly` model. In contrast, `SimPONet` handles this effectively, as it benefits from additional training on real samples.
> >
> >
> >
> > > **Compare with pre-T Z**
> >
> > We have included this experiment in Section 5.7. Our observations indicate that the CATE error is lowest when the baselines are trained on pre-T Z, and highest when they are trained on post-T X. In contrast, training the baselines using the simulated causal representations results in a middle ground, though it still falls short of the ideal pre-T CATE error achievable. This experiment further highlights that the simulator used in our work is not ideal.
> >
> >
> > > **Add a footnote**
> >
> > We added a footnote on page 11. Thanks for this suggestion.
> >
> > > **Small sample size and augment the simulated samples.**
> >
> > We have added this experiment in Appendix A.8. We make the following observations from the experiment:
> >
> > - Robustness in Low-Data Regimes: SimPONet consistently achieves the lowest CATE error with a significant margin over the baselines when training data is limited (10% and 25% of the full training size), showcasing its robustness in low-data settings.
> >
> > - Sustained Performance with Increasing Data: While baseline methods improve with more training data (50% and 75%), SimPONet maintains comparable performance to the baselines at 50% training and achieves further error reduction at 75%, outperforming many baselines that plateau after 50% training size.
> >
> >
> > > **Test with a worst case simulator.**
> >
> > We included this experiment in Sec. 5.6, where we varied the quality of the simulator by adjusting $\gamma_\tau$. A value of $\gamma_\tau = 1$ represents the worst-case simulator. Overall, we observe that the SimPONet’s factual loss component makes it contain the CATE error.

---

### Review · Reviewer_6Vao · 2024-11-15

**Summary Of Contributions:**

The authors tackle the challenging problem of CATE estimation when only post-treatment covariates are available. Identifiability is not possible in this situation, since a backdoor path exists between treatment and outcome. They propose a solution SimPONet which involves the simulation of counterfactual data, and in turn they derive a generalisation bound showing how simulated data can affect CATE estimation error as a function of the degree of discrepancy between the simulated and real data. They show the degree to which the simulated data improves on observational data alone when being used to estimate the CATE, and provide a range of interesting related results to highlight, for example, the impact on the CATE error when only the post-treatment covariates are used.

**Audience:**

Yes

**Broader Impact Concerns:**

None.

**Claims And Evidence:**

Yes

**Requested Changes:**

It’s 90% clear to me what the numbers are in the results. e.g. for 2.82(0.27) in Table 1, so 0.27 is the p-value, and 2.82 is  the mean-squared error on the CATE? Maybe just add some words in the relevant table and/or figure  caption to confirm this would help. If the authors could also briefly explain the colour coding in the captions this would also increase clarify.

Finally, I wonder if I misunderstood something, but if the method enables one to infer the pre-treatment Z and this can be used in other methods, I feel the authors could make a bigger emphasis of this strength - such a pre-treatment Z could be useful for other downstream tasks, perhaps?

I’m not sure why related work is in the appendix. This is the opportunity to situate the contribution with respect to other peoples’ works, and also to build a strong motivation for what is missing in this prior work. Maybe there is, at least, a compromise to be made here.

**Strengths And Weaknesses:**

*Overall Strengths*

The paper is clearly written (with a couple of minor exceptions, see below) and the idea is well-motivated and presented. The implementation of the idea itself seems sound and I appreciated the investigations concerning the use of simulated data in the context of CATE estimation. The authors evaluated on multiple datasets, including the well-known IHDP and ACIC benchmark variants, and highlight that the approach, as well as the variations of the approach, have results with are consistent with expectations and well explained. Furthermore, the authors have associated code which includes the evaluations. I think this is especially valuable given that evaluation of causal methods is somewhat variable (not falling into the traditional supervised learning paradigm) - in my view it is not enough to say which datasets were evaluated on with what metrics, and there are often details which one only discovers by reading the code itself.

*Some other miscellaneous positives:*

The presentation of the three approaches in section 4 was appreciated - added a good intuition for the relative strengths and weaknesses before presenting SimPONet.

For RQ2 I think the use of normalising flows is a good choice in this instance.

*Some questions / points of discussion:*

Relatively minor point, and I’m not confident about the best solution, but a t-test seems inappropriate given the shared variance arising from the use of the same datasets for evaluation. I think this is talked about here: https://ieeexplore.ieee.org/document/6790639 and some solutions are also offered. Interested to hear the opinion of the authors here. I think the McNemar’s test is OK for paired comparisons, and possibly applicable here.  In any case, if the authors wish to take the differences between results seriously, it might be worth exploring the options a bit.

On a related (also minor) note, and I  very well might’ve understood something here, but you say that you ran a one-sided t-test, with SimPONet as the baseline where low p values favour SimPONet.  I just want to clarify that this presumably means that if the effect size of the comparison method is in the opposite direction to that which the test assumes, the p-value will be high even though the method is worse than SimPONet.

I was also wondering whether the authors can speak to any possible consequences of assuming Y is drawn from N(.,.).

*Some Weaknesses*

I think my biggest concern relates to the benchmarks / evaluations. To clarify, this is not a serious weakness, and I feel like they are sufficiently convincing, but could be stronger. I believe Curth et al.,  discussed the limitations associated with these benchmarks in the same paper the authors reference with regards to the use of standard deviation metrics.  Sadly I don’t have a good alternative, except for adding more of them and thereby ascertaining some kind of consistency. I think the field is generally lacking a good solution here. Perhaps the authors could comment on why, for instance, they might or might not consider Jobs and Twins, for instance (I can think of good and bad attributes thereof).

---

> ### Author Response · Authors · 2024-11-27
> **Rebuttal Response**
>
> Thanks so much for appreciating our work and many insightful questions. Please see our responses below.
>
> > **Q1: Shared Variance**
>
> Thank you for highlighting this point and linking the relevant paper. We try to clarify with an example: consider two algorithms $A$ and $B$ trained on $n$ datasets, resulting in paired errors $\{(e_A^1, e_B^1), \dots, (e_A^n, e_B^n)\}$. The referenced paper says that shared variance becomes a concern when the differences $d^i = e_A^i - e_B^i$ are correlated. This is often the case because the $n$ datasets used for training the algorithms $A, B$ are obtained by sampling with replacement from the original training dataset leading to overlapping training training subsets. The paper particularly highlights that this issue is more pronounced in scenarios with small training data, where sampling with replacement causes higher overlap percentages.
>
> However, this is not a concern in our paper because each dataset instance $i$ corresponds to an entirely different dataset. For synthetic datasets, both the real and synthetic data-generating processes (DGPs) change with each seed. For IHDP, each seed represents a unique DGP. Similarly, the ACIC dataset consists of 77 seeds, where each seed represents a distinct dataset. This makes the overlap percentage as 0 for these datasets.
>
> For ACIC-2, 7, 26, to compute error bars within specific seeds (e.g., 2, 7, 26), we ran experiments across 5 seeds. These real training datasets were obtained by sampling with replacement, but the synthetic dataset still changed, which influences the simulated representations learned. Hence, there can be an overlap on the real dataset but the overlap on the synthetic dataset is still 0. Additionally, ACIC datasets are large, containing around 10,000 samples. Thus, we believe that the shared variance problem is perhaps not a problem here.
>
> We chose to retain $p$-values for the $t$-test to remain consistent with prior CATE works (e.g., Curth et al., PairNet). However, if the reviewer finds McNemar's test more appropriate, we are open to replacing it.
>
> > **Q2: Explaining the $t$-Test**
>
> Please refer to our response to Q1. We evaluate the statistical significance of the differences $d^i = e_A^i - e_B^i$. Here, we fix SimPONet as the reference algorithm $A$ across all entries in the table and vary $B$ across baselines. A one-sided paired $t$-test is then conducted to determine whether $d^i < 0$.
>
> We also request the reviewer to take a look at the code that performs the t-test, available in the second cell of the following anonymized notebook: [collate_csvs_post.ipynb](https://anonymous.4open.science/r/catenets-simponet/collate_csvs_post.ipynb).
>
> > **Q3: Assuming $Y$ is Drawn From $\mathcal{N}$**
>
> We assumed the observed outcomes $Y$ are drawn from a Gaussian distribution with a mean of $\mu_t(z)$ and variance $\sigma_y^2$. This variance reflects irreducible noise that cannot be minimized through training. A Gaussian distribution is appropriate because it ensures $E[Y] = \mu_t(z)$, which is our target for approximation. Additionally, with Gaussian, the maximum likelihood training objective naturally leads to the mean squared error loss functions used for training.
>
> > **Q4: Jobs and Twins**
>
> We did not consider the Twins and Jobs datasets because the Jobs dataset is not considered by prior works using the CATENets library. Further for the twins dataset, prior methods generally observe that the CATE baselines typically exhibit little variability in CATE errors, making IHDP and ACIC more suitable for benchmarking.
>
> For example, see:
> - [Dynamic Inter-treatment Information Sharing, Table 1](https://arxiv.org/html/2305.15984v3#S5)
> - [PairNet Table 1](https://arxiv.org/pdf/2406.03864)
>
> To address the reviewer's concern, we have added results with all ACIC dataset seeds in Table 2.
>
> > **Q5: Interpreting Table Numbers**
>
> Yes, the reviewer's interpretation is correct. For an entry such as $2.82(0.27)$, the $p$-value (0.27) indicates the statistical significance. Since this value is relatively large, we cannot conclude with confidence that SimPONet outperforms the corresponding baseline. The number $2.82$ represents the CATE error of the baseline. We have added a footnote on page 11 to this effect.
>
> > **Q6: Experiment with SimPONet’s $Z$**
>
> Thank you for this valuable suggestion. We have incorporated this experiment in Section 5.8. Our observations indicate that SimPONet’s causal representations $Z$ outperform the simulated causal representations learned from the simulator dataset. Further we hope that the reviewer would appreciate the experiment we conducted in Sec. 5.7.
>
>
> > **Q7: Related Work in the Appendix**
>
> Thanks for this suggestion. We have moved the Related Work Sec. 2 in the main paper. Further, we hope that the reviewer would appreciate Sec. 2.3 where we provide real world examples of using simulators for estimating CATE.

---

### Comment · Action_Editor_2ABW · 2025-01-20
**Camera-ready issue**

Dear authors,

There appears to be an error with the margins in your camera-ready version. They are quite a lot larger than the style file's margins.
Please resubmit a version that conforms to the template.

Best,
AE

---

> ### Author Response · Authors · 2025-01-22
> **Revised camera ready paper uploaded.**
>
> Dear AC,
>
> Apologies for the formatting issues. We had included `geometry` package and that inadvertently caused the margin issue.  We have corrected the issue and uploaded the revised version of the paper. Thanks very much.

---

### Decision · Action_Editor_2ABW · 2024-12-20

**Recommendation:** Accept as is

**Comment:**

The reviewers were unanimous in leaning toward acceptance of this work. All appreciated the novelty of the formulation and approach, and the revisions made to the manuscript following the review.

**Audience:**

Yes, there is a clear TMLR audience for this work.

**Claims And Evidence:**

The reviewers identified no issues with unsupported claims after the revision.